# Photoacoustic computed tomography monitors cerebrospinal fluid dynamics and glymphatic function

Seongwook Choi [1,5], Jiwoong Kim[1,5], Hyunseo Jeon[1,5], Yejin Lee[2,3], Gyeongju Lim[2], Won-Min Ju[2], Kyuwan Kim [4], Dong Soo Lee[1], Yun-Sang Lee [2,3], Jae Sung Lee [2,4], Gi Jeong Cheon [2,3], Yoori Choi [2,3] ✉ & Chulhong Kim [1] ✉

Cerebrospinal fluid (CSF) continuously circulates through the brain and surrounding tissues to remove metabolic waste, a process that becomes less efficient with ageing and in neurodegenerative disease. Visualizing this drainage in living animals has been difficult because existing imaging tools either lack depth, require radioactive tracers, or are too slow to capture dynamic flow. Here, we show that whole-body photoacoustic computed tomography (PACT) enables three-dimensional, real-time tracking of CSF transport in mice using indocyanine green. We visualize CSF movement from the spinal canal into the brain, quantify its efflux under different anesthesia conditions, and detect impaired clearance in a mouse model of Alzheimer's disease. Compared with healthy animals, diseased mice retain the tracer in the brain for several days, indicating reduced waste removal. These results establish PACT as a non-invasive platform for measuring CSF and glymphatic function in vivo, providing a way to study how brain fluid transport is altered in ageing and neurological disorders.

Cerebrospinal fluid (CSF) in rodent model both nourishes brain cells and clears the central nervous system (CNS) of metabolic waste and pathological proteins such as amyloid beta (Aβ) and tau, which cause neurodegenerative diseases[1,2]. Because the brain lacks a general lymphatic system, waste is cleared via CSF circulation in the glymphatic system, a specialized perivascular network that promotes the exchange of CSF and interstitial fluid (ISF), facilitating the removal of metabolic waste and neurotoxic proteins. Passing through the subarachnoid space, CSF flows into the deep brain, penetrating the brain parenchyma and mixing with the ISF[3–5]. The combined CSF/ISF fluid clears waste and drains via arachnoid granulations into the venous sinuses and via meningeal lymphatic vessels into the cervical lymph nodes[6,7]. The meningeal lymphatic system disposes of brain and spinal cord waste by transporting ISF and CSF[1,8,9].

Recent studies have revealed that the meningeal lymphatic vessels exhibit age-related alterations, which are associated with impaired CSF clearance and an increased risk of neurodegenerative diseases[6,10]. These age-related changes include a decrease in the diameter and coverage of meningeal lymphatic vessels, as well as reduced drainage of CSF macromolecules into the deep cervical lymph nodes. The impairment of meningeal lymphatic function with age leads to a slowdown in the paravascular influx of CSF macromolecules and the efflux of ISF macromolecules, potentially contributing to cognitive decline.

[1]Department of Electrical Engineering, Convergence IT Engineering, Medical Science and Engineering, Institute of Artificial Intelligence, and Medical Device Innovation Center, Pohang University of Science and Technology, Pohang, Republic of Korea. [2]Department of Nuclear Medicine, Seoul National University Hospital, Seoul, Republic of Korea. [3]Department of Molecular Medicine and Biopharmaceutical Sciences, Graduate School of Convergence Science and Technology, College of Medicine or College of Pharmacy, Seoul National University, Seoul, Republic of Korea. [4]Department of Nuclear Medicine, College of Medicine, Seoul National University, Seoul, Republic of Korea. [5]These authors contributed equally: Seongwook Choi, Jiwoong Kim, Hyunseo Jeon. ✉e-mail: yns086@snu.ac.kr; chulhong@postech.edu

Further, dysfunction of the meningeal lymphatic vessels has been implicated in the pathology of Alzheimer's disease (AD). In transgenic AD mouse models, disruption of meningeal lymphatic vessels promotes Aβ deposition in the meninges, closely resembling human meningeal pathology, and aggravates parenchymal Aβ accumulation[11]. These findings suggest that meningeal lymphatic dysfunction may be an aggravating factor in AD pathology and in age-associated cognitive decline[12].

For these reasons, glymphatic and lymphatic imaging is invaluable in visualizing the CSF route and evaluating the lymphatic function by assessing the CSF clearance rate. Understanding the drainage pathways and their age-related changes can guide strategies for enhancing meningeal lymphatic function to prevent or delay neurological diseases. However, as emphasized in a recent comprehensive review[13], rodent glymphatic transport cannot be directly extrapolated to human physiology, as the underlying mechanisms operate on fundamentally different spatial and temporal scales. Nevertheless, rodent models provide critical mechanistic insight into compartment-specific CSF and lymphatic processes, and they remain indispensable for dissecting these pathways using diverse experimental and imaging tools. This underscores the need for appropriate modalities capable of resolving CSF movement, lymphatic drainage, and waste-clearance dynamics with sufficient spatial and temporal resolution.

To visualize the glymphatic system, fluorescence imaging (FLI), magnetic resonance imaging (MRI), and positron emission tomography (PET) are widely used. FLI measures the dynamics of CSF flow by detecting the signals from fluorescent contrast agents injected into the CSF-filled subarachnoid space. Generally, near-infrared (NIR) fluorescent dyes such as indocyanine green (ICG) are used to visualize CSF's distribution along the neural axis and its flow to lymph nodes, and to analyze lymphatic contractility during circulation[14–16]. MRI, including dynamic contrast-enhancement MRI (DCE-MRI), is also widely used to visualize CSF flow and monitor its drainage pathways, typically requiring gadolinium-based contrast agents. DCE-MRI can trace the CSF flow from the subarachnoid space, including the cisterna magna (CM), along basal meningeal lymphatic vessels to the cervical lymph nodes (cLNs), and can assess differences in CSF clearance efficiency that depend on the type of anesthetic used[17,18]. PET, based on the radioactivity of nuclide-labeled tracers, allows more sensitive visual and quantitative analysis of CSF lymphatic drainage. Recently, [$^{64}$Cu] Cu-albumin PET protocols have clearly visualized the cLNs and sacral lymph nodes (sLNs) to evaluate the drainage of intrathecally injected tracers[19]. Further, by comparing lymphatic function between adult mice (6 months) and aged mice (15 months), PET successfully demonstrated higher CSF retention in the aged mice[19]. However, these conventional imaging methods have inherent limitations. FLI lacks depth information and the spatial resolution of in vivo imaging, restricting 3-D analysis for accurate tracer monitoring. MRI can provide high-resolution 3-D imaging but is constrained by its long acquisition times and high operational costs, making it less practical for tracking CSF dynamics. PET lacks detailed anatomical information and relies on radioactive isotopes, raising safety concerns. Additionally, PET requires multimodal approaches, such as X-ray computed tomography or MRI, to compensate for its limited anatomical resolution. The ideal imaging modality for tracking CSF would provide high resolution, a high frame rate, and comprehensive 3D structural information.

Photoacoustic computed tomography (PACT), an emerging imaging technique, captures 3-D structural and molecular information with high spatiotemporal resolution[20–24]. When laser pulses illuminate a tissue, it absorbs energy and undergoes thermal expansion, generating photoacoustic (PA) signals in the form of ultrasound (US) waves[25]. Following this expansion, the tissue cools down and contracts through thermal diffusion before the next laser pulse arrives. This expansion-contraction cycle repeats periodically at the laser pulse repetition rate, producing continuous PA signals, with temperature fluctuations typically in the micro to milli-degree range[26–28]. This technique capitalizes on the rich optical contrast of biomolecules, particularly endogenous chromophores like hemoglobin, to generate detailed visualizations of anatomical structures without the need for external contrast agents[29–32]. PACT, unlike PET, can also provide molecular imaging using a non-radioactive exogenous contrast agent[19,32]. PACT's ability to both image anatomical structures endogenously and track exogenous agents makes it a groundbreaking alternative to other imaging modalities in glymphatic/lymphatic imaging. PACT also enables real-time CSF dynamics monitoring with high temporal resolution.

In this work, we demonstrate the capability of PACT as a murine glymphatic/lymphatic imaging technique, using ICG as a tracer to monitor and assess CSF dynamics and glymphatic function. First, we demonstrate that whole-body PACT can effectively visualize CSF dynamics in key lymphatic regions, including the subarachnoid space adjacent to the spinal cord, as well as the CM. This capability highlights PACT's potential to provide functional insights into CSF drainage pathways. Second, by comparing the real-time CSF efflux of mice anesthetized with ketamine/xylazine (K/X) and those anesthetized with isoflurane (ISO), we confirm that PACT's real-time imaging capability enables the analysis of CSF flow dynamics in brain regions such as the transverse sinus and CM. Finally, we employ whole-body PACT to evaluate long-term CSF drainage by confirming impaired brain waste clearance in AD transgenic mice.

This study highlights the technical prowess of PACT in glymphatic/lymphatic system imaging and underscores its potential to advance CNS research. By providing an efficient method with high spatio-temporal resolutions for studying CSF dynamics in the glymphatic/lymphatic system, PACT paves the way for non-invasive imaging.

## Results

### In vivo CSF PA whole-body monitoring

To demonstrate the CSF assessment capabilities of the PACT system, we used a previously developed PACT system detailed in our previous study (Fig. 1a)[20,31]. Figure 1b shows a maximum amplitude projection (MAP) whole-body image of the dorsal side of a mouse, in which various organs and vascular structures can be identified[20,31]: 1, transverse sinuses; 2, brown adipose tissue; 3, rib; 4, spleen; 5, spinal cord; 6, intestine; 7, kidney; and 8, common iliac artery. Figure 1c shows a depth-encoded PACT image with a penetration depth of approximately 9 mm.

Using the PACT system, we monitored the mouse's whole body for 24 h following an intrathecal (IT) injection of 8 μL of 1 mM ICG over 10 min (Fig. 1a). Note, to demonstrate a broader application of PACT-based CSF assessment, we additionally observed signal increase and removal in the CM following direct CM injection, as well as signal enhancement in the sentinel lymph node (SLN) after intradermal injection into the forefoot. (Supplementary Fig. 1)[33] PACT imaging was performed pre-injection, and subsequently at 30 min and 24 h after finishing the injection (See Methods). Upon IT injection, ICG was rapidly distributed into the CSF in the subarachnoid space and moved along the spinal cord in the rostral and caudal directions, serving as an indicator of CSF flow[34]. Before CSF in the spinal cord flows into the brain, it necessarily passes through the CM, a CSF-filled space located between the posterior surface of the medulla oblongata and the underside of the cerebellum[35–37]. Since the PA signal from the ICG in the CM allows observation of the flow of CSF, we analyzed the signals in the spinal cord and CM. To accurately analyze the ICG signal and precisely track the CSF dynamics, we performed blind unmixing to extract ICG signals in whole-body images captured using 4-wavelength illumination (756, 780, 796, and 900 nm). This technique is based on the distinctly different light absorption characteristics of ICG and various endogenous

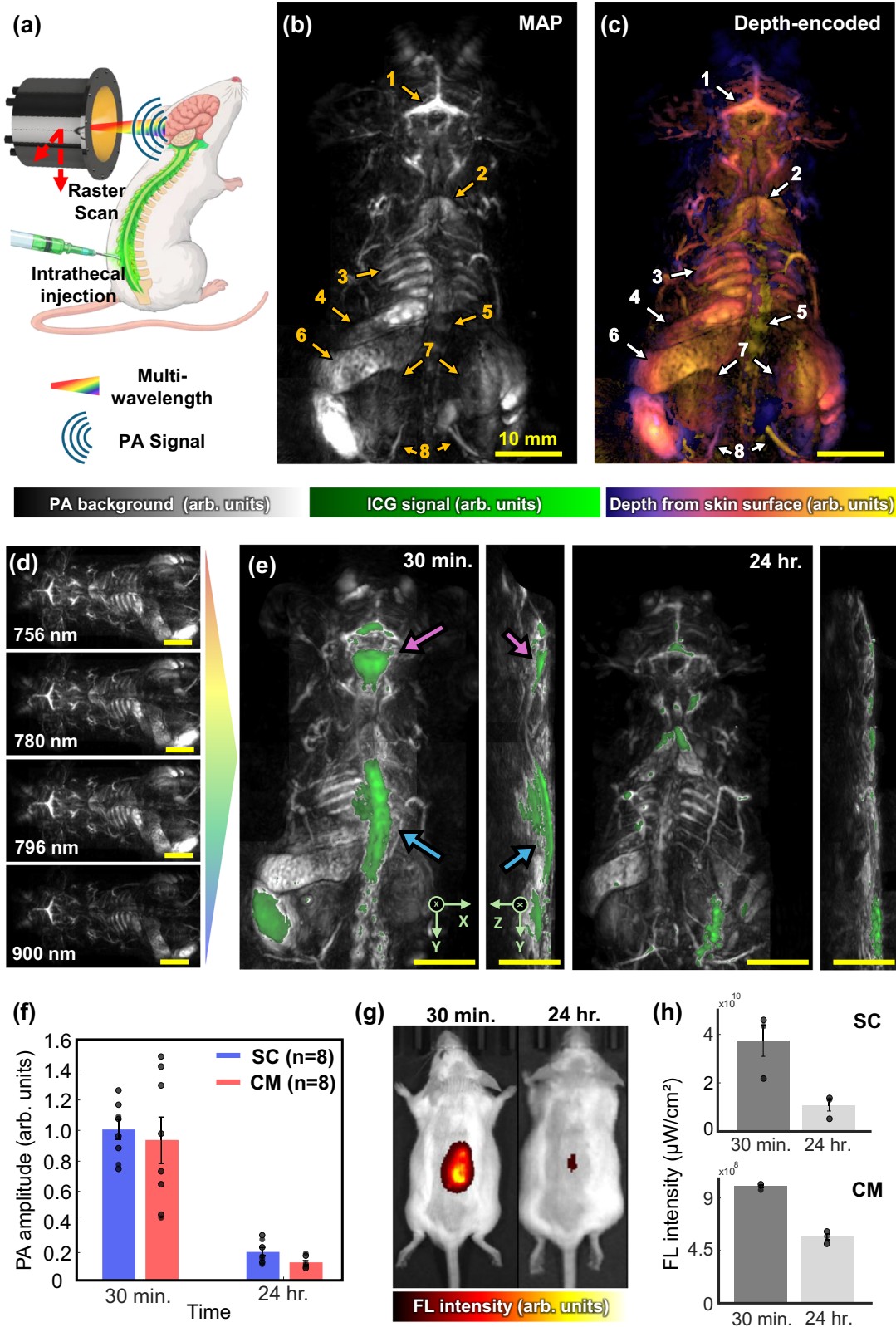

backgrounds (e.g. hemoglobin, lipid, and water) (See Methods)[38–40] (Fig. 1d and Supplementary Fig. 2). Figure 1d shows multispectral PACT images captured prior to IT injection that highlight different PA intensities in various organs, such as the spleen, ribs, and intestines. Figure 1e shows whole-body MAP images in the XY and YZ plane at 30 min and 24 h after IT injection (Supplementary Video 1). For visual comparison, see the pre-injection whole-body image in

Fig. 1b (Supplementary Video 2). The background images (gray) were acquired at a wavelength of 796 nm, while the unmixed ICG signal intensity is overlaid in green. At 30 min after injection, a significant increase in the ICG signals is seen in the subarachnoid space at the spinal cord (sky blue arrows) and CM (pink arrows), capturing the flow of the injected ICG with the CSF from the subarachnoid space at the spinal cord to the CM. At 24 h after injection, the ICG signal in

**Fig. 1 | PACT monitoring of whole-body CSF dynamics. a** Schematic of PA imaging and IT injection. **b** In vivo MAP and **c** depth-encoded PACT images of a mouse in the dorsal plane. **d** Multispectral whole-body PACT images at 756, 780, 796, and 900 nm wavelengths. **e** MAP structure images (gray colormap) and spectrally unmixed ICG map (green colormap) in the XY and YZ plane taken at 30 min and at 24 h after IT injection of ICG. The pink arrow indicates the cisterna magna (CM), and the sky blue arrow indicates the spinal cord (SC). **f** Changes in PA signals over time in the SC ($n = 8$) and CM ($n = 8$) in the ICG unmixed map. **g** In vivo FL images obtained at 30 min and 24 h following IT injection. **h** Changes in FL signals over time in the SC ($n = 3$) and CM ($n = 3$). CSF cerebrospinal fluid, PACT photoacoustic computed tomography, PA photoacoustic, MAP maximum amplitude projection, ICG indocyanine green, IT intrathecal, FL fluorescence, SC spinal cord, CM cisterna magna, arb. units, arbitrary units; 1, transverse sinuses; 2, brown adipose tissue; 3, rib; 4, spleen; 5, spinal cord; 6, intestine; 7, kidney; 8, common iliac artery; All statistical data are presented as mean ± standard errors, and individual data are overlaid as dot plots.

each region has returned to pre-injection levels, indicating that the ICG has been cleared by the flow of CSF[41]. Additionally, Supplementary Fig. 3 visualizes the cranial CSF lymphatic drainage pathway, providing physiological insight into CSF clearance. The meningeal lymphatic vessels are distributed along the transverse sinus and superior sagittal sinus, and this route is visualized in Supplementary Fig. 3. ICG was cleared along the meningeal surface surrounding the brain through the meningeal lymphatic vessels (red arrows), consistent with previously reported lymphatic flow pathways in mice[42]. Note that rodents lack arachnoid granulations; therefore, their cranial CSF efflux relies primarily on lymphatic and perineural pathways, such as meningeal lymphatics, cranial and spinal nerve sheaths and nasopharyngeal lymphatic plexus as previous literature has reported[43–47]. Due to the limited molecular sensitivity of PACT, our imaging primarily captures the dominant meningeal lymphatic pathway, and further details regarding these limitations are provided in the Discussion section. These results demonstrate that PACT, using ICG as a CSF tracer, can visualize the CSF flow in the subarachnoid space. Note that the ICG signal observed in the intestine in Fig. 1e is influenced by artifacts. These artifacts arise from feces in the intestine (purple arrow), causing saturation of the digitized PA signal. As a result, for this kind of strong absorbers, the PA signal at each wavelength does not accurately represent the absorption spectrum of the material components, and the artifact occurs. To quantitatively analyze the changes, the average PA signal was calculated by manually designating regions-of-interest (ROIs) in the spinal cord region and CM in the XY-plane unmixed images (See Methods). Figure 1f shows the different ICG signal levels in the spinal cord and CM at 30 min and 24 h after IT injection ($n = 8$). The 30-minute CM value is nearly the same as the PA amplitude of the spinal cord, showing that a significant amount of ICG moved from the spinal cord to the CM right after injection (spinal cord, $1.0 \pm 0.06$ arb. units; CM, $0.93 \pm 0.14$ arb. units). At 24 hours post-injection, these values have decreased to $0.19 \pm 0.02$ arb. units in the spinal cord and $0.12 \pm 0.01$ arb. units in the CM, confirming that the ICG has been cleared through CSF circulation ($t(7) = 14.08$, $p = 2.2 \times 10^{-6}$, effect size statistic ($d$) = 4.98, 95% Confidence Intervals (CI) = [0.67 0.94] for the SC and $t(7) = 5.00$, $p = 0.0016$, $d = 1.77$, 95% CI = [0.43 1.19] for CM, comparing 30 min and 24 h time points).

We used in vivo fluorescence imaging to validate the movement of the CSF tracer observed after IT injection in PACT whole-body images (Fig. 1g). We acquired dorsal in vivo images at 30 minutes and 24 hours after IT injection of ICG, using the same method as for PACT ($n = 3$). Consistent with the PACT results, robust FL signals were detected in the spinal cord ($37 \pm 6.2 \times 10^9$ μW/cm²) and CM ($100 \pm 1.1 \times 10^7$ μW/cm²) at 30 minutes after injection, but their levels exhibited a marked decrease at the 24-hour time point ($12 \pm 2.2 \times 10^9$ μW/cm² for the spinal cord and $57 \pm 2.5 \times 10^7$ μW/cm² for the CM, Fig. 1h). To more clearly visualize the FL signal in the CM region, Supplementary Fig. 4 presents an enlarged view of the CM with an adjusted color scale. It was confirmed that the FL signals in the CM had completely disappeared at 24 h in the images displayed using the same colormap. By validating the whole-body tracking of the CSF tracer using fluorescence imaging, we confirmed the capability of PACT to trace CSF flow within the subarachnoid space in vivo, while simultaneously capturing morphological details.

## Real-time CSF dynamics monitoring

To demonstrate PACT's real-time monitoring capability, we monitored CSF dynamics in vivo. Raster scanning would not allow us to observe how CSF behaves in real-time in whole-body images because of the scanning time. Thus, to capture the CSF dynamics without raster scanning, we limited the ROI to the brain region and CM for 30 minutes after IT injection of ICG. To accommodate the channel limitation of the data acquisition board (DAQ) and to maximize imaging speed, PACT images were acquired using only an 800 nm wavelength, but even without spectral unmixing the ICG signal increase could still be visualized by subtracting the pre-infusion image from the post-infusion image (See Methods). To demonstrate the capture of CSF drainage dynamics using PACT, we compared the efflux from the CM between mice anesthetized with K/X and those anesthetized with ISO. In Fig. 2a, the XY-MAP images of the mouse brain show a higher signal increase in the CM (orange arrows) under K/X anesthesia than they do under ISO (the real-time video is available in Supplementary Video 2). The K/X anesthetized group shows a significantly increased CM signal visualized 5 minutes after the start of the injection, while the ISO group shows a lower signal increase even after a 10-minute time point. The CM's strong signal increase is also observed in the YZ-MAP, showing a significant difference in the signal levels between K/X and ISO. Figure 2b displays the increase in signal intensity in the CM region over the 30 min following the injection compared to the 0-minute signal. The shaded areas in Fig. 2b represent the standard error ($n = 8$) of each group, and the green area indicates the injection period. The quantified PA signal intensity in the CM region at the 30 min point shows a significant difference between the two anesthetic groups, with K/X anesthetized mice exhibiting a $4.37 \pm 0.86$ arb. units increase from pre-injection, compared to the ISO anesthetized group's $2.15 \pm 0.20$ arb. units increase ($t(14) = 2.36$, $p = 0.033$, $d = 1.18$, 95% CI = [1.21 3.24]). These results are consistent with previous studies indicating that the CSF movement is significantly greater in mice anesthetized with K/X compared to those anesthetized with ISO[16,17,48]. Note that this phenomenon is associated with K/X anesthetized mice that exhibit higher EEG delta power than ISO anesthetized mice. Our PACT-based real-time analysis of anesthetics' impact on CSF movement aptly demonstrates PACT's ability to assess CSF dynamics.

Figure 2c presents the validation of these results through FL imaging. We compared the FL signal intensity induced by ICG accumulation in the CM at 30 min post-IT injection in mice anesthetized with ISO and K/X ($n = 3$). In the ISO group, a FL signal of $(2.3 \pm 0.72) \times 10^9$ μW/cm² was observed in the CM region 30 min post-injection, whereas in the K/X group, a signal of $(10.8 \pm 0.83) \times 10^9$ μW/cm² was detected ($t(4) = 6.31$, $p = 0.003$, $d = 5.15$, 95% CI = [$4.8 \times 10^9$ $1.2 \times 10^{10}$]), which corresponds to approximately 4.7 times higher ICG signal intensity. Supplementary Fig. 5 shows FL images of the K/X and ISO group's CM and brain region at 30 minutes post-injection. The FL images also confirmed that the K/X group exhibited stronger FL signals in the CM at 30 min, which is consistent with the PACT results. In conclusion, these findings underscore the capability of PACT to dynamically monitor CSF flow under varying physiological conditions. By demonstrating significant differences in CSF movement influenced by anesthetic types, PACT proved to be a valuable tool for real-time assessment of CSF dynamics and related brain mechanisms.

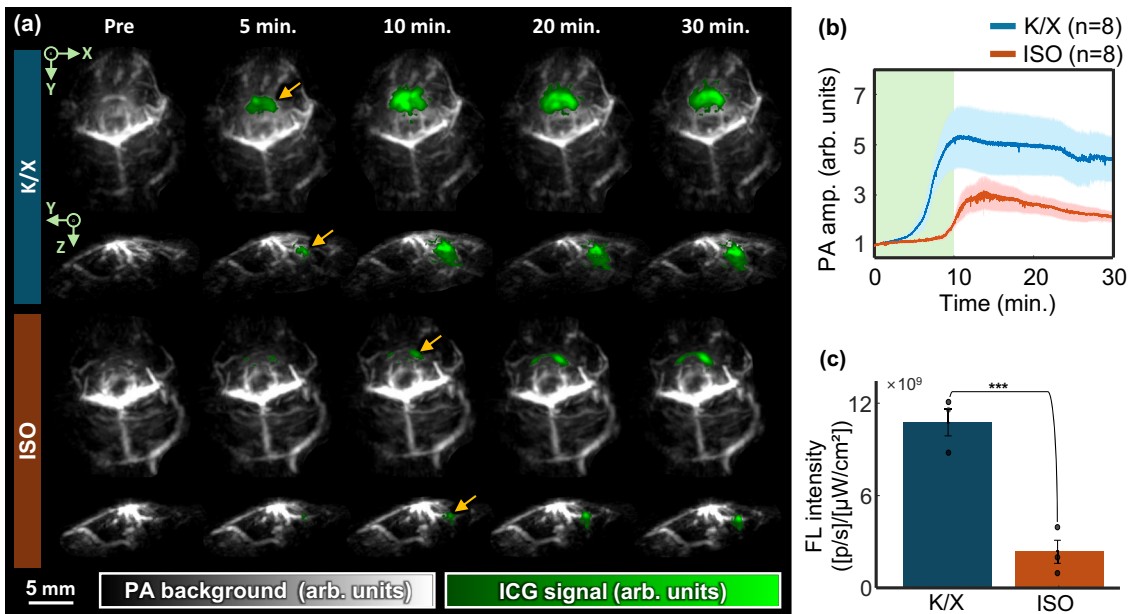

**Fig. 2 | PACT monitoring of brain CSF dynamics. a** Brain PACT images in the XY plane and YZ plane of ketamine/xylazine (K/X) anesthetized mice and isoflurane (ISO) anesthetized mice. Images were captured at pre-injection and 5, 10, 20, and 30 min post-injection of ICG. Orange arrows indicate the CM. **b** Changes in PA signals from the CM over time (K/X: $n = 8$, ISO: $n = 8$). The shaded areas represent the standard error of each group, and the green area indicates the injection period. **c** Comparison of in vivo FL intensities of CM at 30 min post injection of ICG, as measured using the IVIS system (K/X: $n = 3$, ISO: $n = 3$). Comparisons were performed using an independent two-tailed t-test (t(4) = 6.31, $p = 0.003$, d = 5.15, 95% CI = [$4.8 \times 10^9$ $1.2 \times 10^{10}$]). All statistical data are presented as mean ± standard errors, and each data point represents an individual animal. ***$p < 0.005$; CSF cerebrospinal fluid, PACT photoacoustic computed tomography, PA photoacoustic, MAP maximum amplitude projection, ICG indocyanine green, K/X ketamine/xylazine, ISO isoflurane, FL fluorescence, CM cisterna magna, arb. units, arbitrary units; Error bars represent the standard errors.

## Comparison of CSF drainage in mice with Alzheimer's disease and wild type mice

To show PACT's potential for long-term CSF assessment in studying various neurological diseases, we compared the AD and wild-type (WT) mouse models by monitoring ICG clearance from the brain after intraparenchymal injection. Compared to WT mice, AD mice showed impaired parenchymal tracer clearance in association with the accumulation of Aβ, which has been associated with several well-documented alterations in CSF and lymphatic transport. Prior studies have demonstrated reduced peripheral lymphatic pumping and diminished CSF outflow[14], impaired glymphatic influx and tau clearance[49], compromised lymphatic Aβ removal[50], and partial restoration of glymphatic–lymphatic drainage via focused ultrasound[51]. Collectively, these findings indicate that multiple components of the CSF−lymphatic transport system are disrupted in AD, consistent with the impaired tracer clearance observed in our study[13,14,49,50,52]. We intraparenchymally injected 2 µL of 1 mM ICG into the striatum of 5XFAD AD mice and age-matched WT mice over 2 min (See Methods). Whole-body images were obtained pre-injection, and at 1, 6, 24, and 96 h after injection. The ICG signals captured by 4-wavelength PACT were then unmixed. Figure 3a, b show XY- and YZ-MAP images of WT mice and AD mice with overlaid unmixed ICG signals (the backgrounds were acquired by the 796 nm wavelength). At 1 h after injection, dot-shaped ICG signals were detected in the brain parenchyma of both the WT and AD mice (orange arrows in Fig. 3a, b). As seen in the YZ-MAP image below, in which the magnified region is indicated by the orange dotted box, the ICG signals were observed at a depth of approximately 3.8 mm, the point where the ICG was administered. In the WT model, the ICG signal gradually decreased over time, while in the AD model, it remained at a similar level at the 1 h time point. By the 96 hours time point, the ICG signal in the WT mouse images had almost disappeared, but a clear dot shape remained in the AD mouse images

(Supplementary Video 4). Figure 3c displays the differences in ICG clearance visualized in the PACT images ($n = 8$), with the average PA signal intensities calculated from manually set ROIs in the striatum in the unmixed ICG images. Cuboid 3D ROIs were designated for the dot-shaped ICG in the striatum to exclude ICG signals caused by possible material leaked during injection or stereotaxic-surgery injury. To analyze ICG PA signal changes at each time point after injection, the values were normalized to the value at 1 h after injection (See Methods). The ICG signals in the striatum showed slightly different trends in the WT and AD mice after 6 h ($0.85 \pm 0.06$ arb. units in WT and $1.03 \pm 0.08$ arb. units in AD at 6 h. t(14) = -1.74, $p = 0.1$, $d = 0.87$, 95% CI = [-0.30 -0.07]). However, differences were increased at 24 hours ($0.69 \pm 0.07$ arb. units in the WT and $0.94 \pm 0.08$ arb. units in the AD; t(14) = -2.19, $p = 0.046$, $d = 1.09$, 95% CI = [-0.38 -0.13]) and became more evident by 96 h. At the 96 h time point, the ICG signal in AD model mice remained at $1.06 \pm 0.07$ arb. units, which was similar to the 1-hour post-injection value, while in WT mice, it decreased to $0.47 \pm 0.04$ arb. units (t(13) = -6.93, $p = 0.00001$, d = 3.58, 95% CI = [-0.50 -6.93]). To validate this result, we compared the residual ICG signal in brain parenchyma between the two groups through ex vivo FL imaging at 96 hours after injection. Figure 3d shows a significant reduction in FL signals from the brain of a WT mouse compared to those from an AD mouse. Quantitatively, the injection site in the WT group exhibited an FL signal of ($9.44 \pm 0.76$) × $10^8$ µW/cm², whereas the AD group showed ($14.2 \pm 0.33$) × $10^8$ µW/cm² (t(4) = −4.65, $p = 0.0096$, d = 5.77, 95% CI = [$-9.5 \times 10^8$ $-4.9 \times 10^8$]), validating the difference in ICG clearance between the two groups observed through PACT. In conclusion, PACT assessed waste clearance in WT and AD mice morphologically and quantitatively using intraparenchymal ICG injections. These results demonstrate the capability and promise of the PACT system for assessing parenchymal waste clearance and CSF-linked fluid-transport in rodent models.

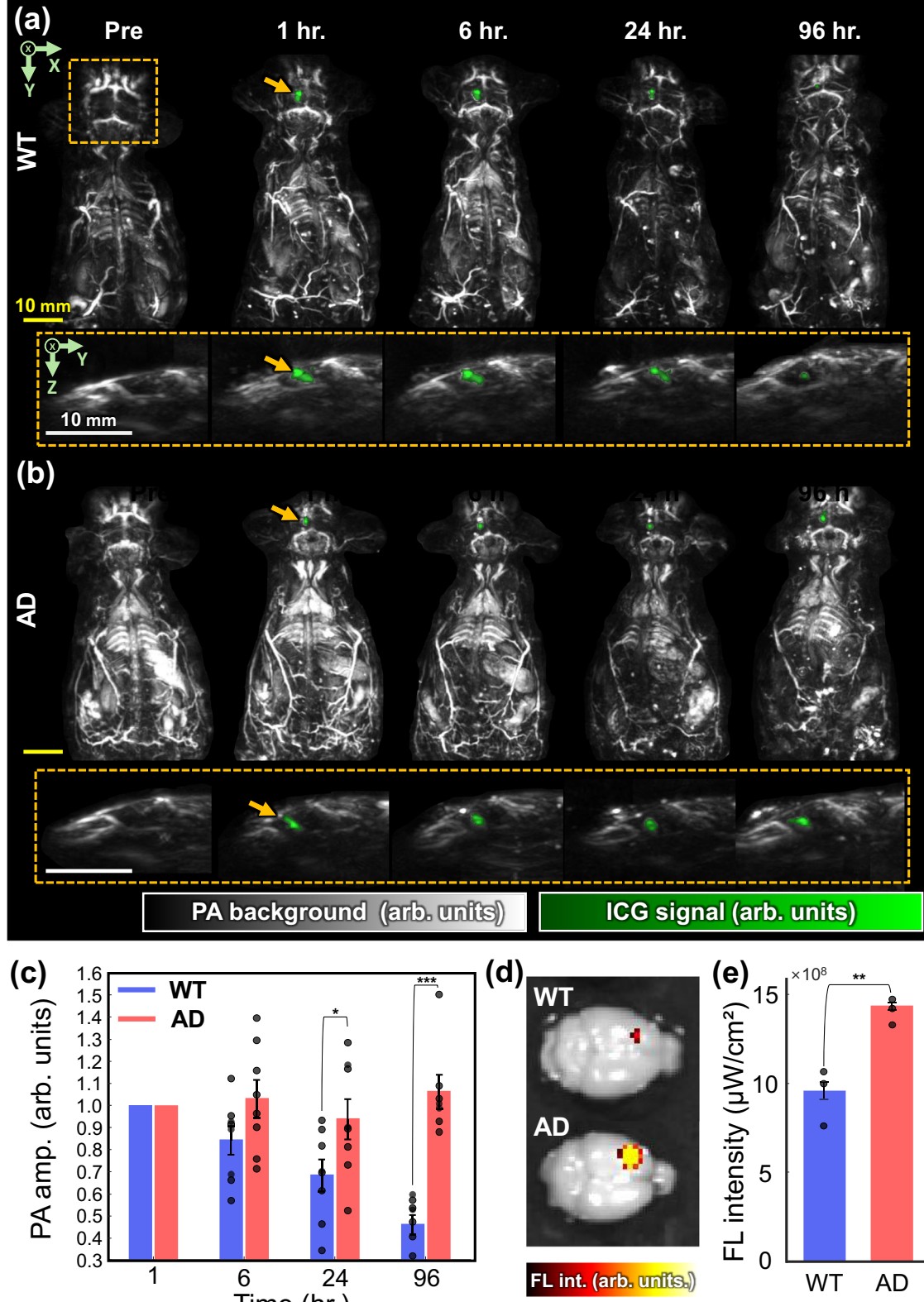

## Discussion

The discovery of the glymphatic-meningeal lymphatic system has revealed that waste clearance mechanisms in the CNS rely on CSF circulation. Therefore, understanding the CSF dynamics in the subarachnoid spaces, as well as the CSF drainage via the meningeal lymphatic system, is crucial for understanding neurodegenerative diseases. To date, CSF circulation and lymphatic drainage have primarily been investigated using FLI, MRI, and PET. However, each technique has fundamental drawbacks: FLI lacks depth and 3-D quantitative information, MRI offers limited temporal resolution and requires lengthy acquisitions, and PET relies on radioactive tracers and lacks detailed anatomical context. In contrast, PACT is an emerging biomedical imaging modality that leverages its multiparametric capabilities and has been widely applied in preclinical mouse studies

**Fig. 3 | PACT monitoring of CSF drainage in AD and WT mice.** PACT whole-body MAP images in the XY and YZ planes of **a** WT mice and **b** AD model mice. Images were captured at pre-injection and 1, 6, 24, and 96 hours after injection of ICG. The orange dashed boxes indicate the zoomed-in brain region in the YZ-MAP images, and the orange arrows indicate the injection site in the striatum. **c** Changes over time in the ICG striatum signals in WT mice and AD mice in the ICG unmixed map (WT: $n = 8$, AD: $n = 8$). **d** Ex vivo FL brain images acquired using the IVIS system at 96 h after injection of ICG. **e** Comparison of the ex vivo FL intensities of the striatum between WT mice and AD mice at 96 hours after injection of ICG (WT: $n = 3$, AD: $n = 3$). Comparisons were performed using an independent two-tailed t-test $t(4) = -4.65$, $p = 0.0096$, $d = 5.77$, 95% CI = $[-9.5 \times 10^8 -4.9 \times 10^8]$ $*p < 0.05$; $**p < 0.01$; $***p < 0.005$; CSF, cerebrospinal fluid; PACT, photoacoustic computed tomography; PA, photoacoustic; MAP, maximum amplitude projection; ICG, indocyanine green; AD, Alzheimer's disease; WT, wild type; FL, fluorescence; arb. units, arbitrary units. All statistical data are presented as mean ± standard errors, and individual data are overlaid as dot plots.

across diverse disease models, including cancer and stroke[29,53]. In this study, we demonstrate that PACT can be utilized to monitor CSF dynamics and glymphatic-related function in rodent models, highlighting its potential as a next-generation platform for evaluating fluid transport in the CNS. Meanwhile, photoacoustic microscopy (PAM) is another imaging technique that utilizes the PA effect, similar to PACT, but focuses on micro-scale imaging as opposed to PACT's macro-scale imaging. There have been studies employing PAM to explore multi-wavelength imaging of the dynamic processes of CSF in meningeal lymphatic vessels[42,54]. These studies have contributed valuable data on the localization and movement of CSF. However, they have several limitations. This limitation fundamentally arises from PAM's single-element scanning architecture, which acquires point-by-point A-line signals rather than simultaneously capturing 2D or 3D volumes. First, previous PAM studies primarily provided non-real-time imaging data, limiting the ability to dynamically monitor and quantify CSF flow across various physiological conditions. This limitation makes it difficult to capture rapid physiological changes in homeostasis. In other words, these studies lack the ability to dynamically monitor CSF flow, which is crucial for understanding glymphatic function. Second, previous research has predominantly focused on localized brain imaging, which restricts the understanding of systemic physiological interactions and cannot provide a comprehensive view of CSF distribution in major lymphatics on a macro scale. Consequently, while previous PAM studies have laid important groundwork for CSF imaging using the PA effect, there remains an opportunity to develop more holistic, real-time, and comprehensive insights into CSF dynamics and glymphatic function via PACT. This study demonstrates the advantages of PACT for analyzing CSF dynamics and glymphatic function.

In this study, we visualized major lymphatic regions in key lymphatic regions, including the subarachnoid space adjacent to the spinal cord and the CM, using ICG as a CSF tracer, which allows observation of CSF's circulation. Additionally, by injecting ICG directly into the brain parenchyma to mimic metabolic waste, we evaluated glymphatic clearance, highlighting PACT's ability to assess brain waste removal. First, our PACT system visualized the distribution of ICG in localized CSF-containing regions, including the CM, transverse sinus, and spinal canal, following IT injection. It also enabled tracking of its spatiotemporal changes, providing insight into CSF efflux patterns. Second, we used the real-time monitoring capability of PACT to dynamically analyze CSF movement after IT injection. We compared the CSF efflux into the CM under different anesthesia conditions, specifically between mice anesthetized with K/X and those anesthetized with ISO. Mice anesthetized with K/X exhibited greater CSF efflux over the initial 30 minutes, indicating that the choice of anesthetic can influence CSF flow dynamics. This real-time capability can observe lymphatic contractions by monitoring signal changes in peripheral lymphatic vessels, providing valuable insights for future studies[15]. Third, we evaluated long-term clearance of waste products from the brain parenchyma in AD and WT mice. ICG was directly injected into the brain parenchyma, and its clearance was evaluated by observing the reduction in ICG signals from the brain over 96 hours. In WT mice, the ICG PA signals decreased over time, but in AD mice the signals remained consistent even after 96 hours. Our study demonstrates that PACT can evaluate the brain's waste clearance capacity by visualizing impaired parenchymal waste removal in AD model mice, which is suggestive of glymphatic system dysfunction, highlighting the technique's potential for advancing research in CNS function and neurological disease. Note, a substantial body of evidence now supports the view that AD involves multisystem impairments of CSF transport, including alterations in subarachnoid mixing, perivascular exchange, and meningeal lymphatic outflow. As highlighted in the recent review paper[13], AD pathology cannot be explained by a single circulation loop but instead reflects compartment-specific failures spanning the CSF, perivascular, parenchymal, and lymphatic domains. In this context, demonstrating impaired parenchymal tracer clearance in AD mice is particularly meaningful: our PACT measurements capture a functional signature of reduced efflux capacity that aligns with the known lymphatic and CSF-transport deficits in AD. By capturing these differential clearance profiles in controlled genetic models (including the anesthesia-dependent CSF-transport differences demonstrated earlier), our study demonstrates PACT's utility as a functional assessment tool capable of resolving disease-associated alterations in CSF and solute transport. Furthermore, for clinical translation, PACT's demonstrated efficacy in clinical settings, including its successful use for imaging human lymphatics and brain imaging, and its obtaining FDA approval, shows its potential for human CSF monitoring[22,55].

The CSF studies using the PACT system can be further improved in three major aspects. First, PACT showed lower sensitivity to the tracer than other conventional CSF monitoring imaging modalities (e.g., DCE-MRI and PET). PACT enabled structural and quantitative evaluation of waste clearance by observing the distribution of ICG as a tracer, however, it was unable to detect the meningeal lymphatic vessels and meningeal lymphatic nodes (e.g., deep cLNs, cLNs, iliac lymph nodes, and sLNs), which are the other routes for waste disposal, limiting accurate analysis of the drainage pathways. Because ICG, the dye used in our technique, is a small molecule that does not remain long in the lymph nodes, PACT was challenged to detect it sensitively. Nevertheless, compared to FL imaging, which provides high molecular sensitivity but is restricted by shallow imaging depth, PACT notably offers complementary advantages, including superior penetration depth and structural 3D information that can be correlated with tracer distribution. As shown in Supplementary Fig. 6, depth-separated PACT slice images clearly delineate the spinal cord (blue arrows), CM (red arrows), and the intervening CSF space, allowing detailed assessment of how the tracer propagates through these anatomical compartments in three dimensions (also see Supplementary Video 1 and 2). Notably, because PACT inherently contains background signals from endogenous absorbers such as hemoglobin, which can reduce molecular sensitivity, we applied spectral unmixing to isolate tracer-specific spectral components and minimize interference from endogenous chromophores. These strengths of spectral PACT enabled us to obtain more detailed, spatially resolved insights into CSF clearance dynamics than would be possible with FL alone. With this rationale, we expect that the advanced materials tailored for PACT-based CSF studies will help overcome this molecular sensitivity[56]. In line with this expectation, we intrathecally injected a high concentration of 12 mM ICG, and

we were able to observe additional lymphatic regions, like the cervical lymph nodes and nasal region in the ventral plane (Supplementary Fig. 7). These results collectively illustrate that although PACT has lower molecular sensitivity than FL-based methods, its depth-resolved and organ-specific imaging capabilities offer unique advantages that can be strengthened further through improved contrast materials, ultimately expanding the utility of PACT-based CSF studies for both preclinical and clinical applications. Second, the current unmixing method based on non-negative matrix factorization requires further refinement. It uses data-driven absorption coefficients instead of fixed values as in conventional PACT-based unmixing, which extracts only oxy- and deoxyhemoglobin. Thus, due to the unknown absorption characteristics of ICG administered inside the body, the unmixing performance was limited when the ICG-enhanced region was relatively small within the entire image or when the ICG signal was not significantly stronger than that from surrounding tissues. This limitation can be improved by developing advanced unmixing algorithms[56,57]. Lastly, due to the inherent limitations of PACT, such as its limited penetration depth and bandwidth, some lymphatic vessels and lymph nodes may be missed. This shortcoming can potentially be overcome by developing suitable hardware systems and applying deep learning techniques in further research[58-61].

## Methods

### PACT system

The developed PACT system effectively detects omnidirectionally propagating PA waves using a customized 1024-element hemispherical US transducer array (Japan Probe, Inc., Japan), and it achieves an isotropic spatial resolution of 380 μm[20,31]. The transducer array has a 60 mm radius, a 2.02 MHz center frequency, and a 54% bandwidth. A custom-made fiber bundle (Opotek, Inc., USA) passing through the center hole of the array delivers the laser beam to the target. We specifically tracked the exogenous material signals based on the optical absorption properties of ICG, using a tunable optical parametric oscillator (OPO) laser (PhotoSonus M-20, Ekspla, Inc., Lithuania; 20 Hz repetition frequency; 3–5 ns pulse width). The output wavelength is freely adjustable within the range of 690–1064 nm, and the pulse energies per unit area on the skin surface at each wavelength are well within the safety limits of the American National Standards Institute (ANSI). The 256-channel DAQ system (Vantage 256, Verasonics, Inc., USA) is triggered by the output of the laser system to receive PA signals, digitizing them with programmable amplification of up to 54 dB at a sampling frequency of 8.33 MHz. Because the US transducer array and the DAQ system have different numbers of channels, the data are processed through a 4:1 multiplexing (MUX) board. Therefore, a 3D single volume requires four laser shots, so the effective imaging frame rate is 5 Hz. Each 3D single volume captured in one imaging sequence encompasses dimensions of 12.8 mm × 12.8 mm × 12.8 mm[20]. To generate a whole-body image, a 3-axis gantry motor system (LSQ150A, Zaber, Inc., USA) moves the target to enable raster scanning at multiple positions, and the obtained single-volume images are combined. During imaging, the animal is secured in a customized holder and partially submerged in water within a water tank connected to the transducer. The animal's body temperature was maintained within a stable range of 34 °C–37 °C using a heating bath circulator placed outside the water tank, and monitored using a needle-type thermocouple system (HYP0-33-1-T-G-60-SMP-M and OM-DAQ-USB-2400, Omega Engineering, US).

### Data-driven blind spectral unmixing

Through multispectral imaging based on the various optical absorption characteristics of specific biomolecules, PAI selectively extracts the distribution of specific chromophores, providing functional information about each biomolecule. The linear mixing model (LMM) represents the mixed PA signal intensity **A** of $m$ pixels resulting from $n$

multiple spectral laser emissions, according to the following equation:

$$\mathbf{A} = \mathbf{WH} + \mathbf{N} \tag{1}$$

where **A** is the $n \times m$ matrix of mixed PA signal intensity, **W** is the $n \times l$ matrix of the endmember that contains the spectrum of each chromophore, **H** is the $l \times m$ matrix of the extracted maps of the relative distribution of each chromophore, and **N** is the $n \times m$ matrix of noise (note that noise is not considered in general spectral unmixing)[38]. When extracting the concentration of oxyhemoglobin (HbO) or deoxyhemoglobin (HbR), the matrix **W** is calculated through the least-squares fitting (LSQ) method between the actual data matrix **A** and the **H** matrix, based on the theoretical absorption spectrum. However, in cases where the theoretical spectrum varies significantly depending on the concentration and medium of a substance like ICG, and is also difficult to measure, data-driven blind spectral unmixing is applied to infer the matrices **W** and **H** from the multispectral data. We also applied a non-negative matrix factorization (NNMF)-based blind unmixing method to extract the PA signal of ICG from whole-body PA images, and the matrices were determined by the following equation[38,62]:

$$\mathbf{A} \approx \mathbf{WH}, \tag{2}$$

$$[\mathbf{W}, \mathbf{H}] = \frac{1}{2}||\mathbf{A} - \mathbf{WH}||_F^2 \tag{3}$$

The matrices **W** and **H** are initialized with an initial guess and iteratively updated to minimize the loss function, expressed as the squared Euclidean distance between **A** and **WH**, while maintaining positive element values. We applied this blind unmixing method to an experimentally acquired 4-wavelength (756, 780, 796, and 900 nm) PA multispectral data set, and the extracted endmembers and unmixed maps closely matched the ICG signal trends observed in the unmixed images (Supplementary Fig. 2). These results confirmed the ICG signal extraction capability of our NNMF-based unmixing method, and we then applied it to segment regions containing ICG in the PA whole-body images, obtaining an accurate interpretation of the results. To visualize the unmixed ICG, we overlaid the background PACT image acquired at 796 nm (gray colormap) with the unmixed ICG signals in a green colormap. During this process, regions of interest (such as the brain, CM, and spinal cord) were segmented based on the objectives of each experiment, and the corresponding ICG maps within these ROIs were overlaid.

### Quantification of representative ICG signals

To quantitatively analyze the changes in ICG PA signals, we segmented the interest volume regions by manually defining polygonal regions-of-interest (ROIs). When analyzing the ICG signals in the spinal cord and CM after intrathecal injection of ICG, the ROIs were defined in the XY-MAP images. In contrast, when ICG was directly injected into the brain parenchyma via intracerebroventricular injection, the analysis was based on 3D ROIs to exclude PA signals caused by material leakage during needle removal and thrombosis from the stereotaxic surgery. First, ROIs were defined on the XY-MAP images and secondary ROIs were segmented on the YZ-MAP images of the selected volume, allowing the striatum to be segmented into a 3D rectangular prism. Subsequently, the analysis was conducted exclusively on the ICG signals. To optimally reflect the signals induced by the exogenous material in the statistical analysis, only the top 10-30% of pixels with the highest PA signals within the ROIs were used for quantification to minimize the impact of overshooting and noise within ROIs[31,63]. The average of the PA signals from the selected pixels, calculated by dividing the sum of their PA signals by the number of pixels, was used

as the representative value for the ROIs. These values were then plotted, along with the standard deviations.

## Animals

For the AD model, 6–8-month-old 5XFAD transgenic mice (#034840-JAX) were used. The 5XFAD mouse carries five familial AD mutations in human APP and PS1. Age-matched littermates were used as WT controls. Mice of both sexes were included in the experiments. For the CSF whole-body PACT (Fig. 1) and CSF dynamics experiments (Fig. 2), 2–5 month-old female ICR mice (Hsd:ICR(CD-1®)) were used. All animal experiments were conducted with approval from the Institutional Animal Care and Use Committee (IACUC) at Seoul National University (SNU-230807-3-4) and Pohang University of Science and Technology (POSTECH) (POSTECH-2024-0025-C2-R1). All procedures adhered to relevant guidelines and regulations regarding the care and use of animals for experimental research. Mice were housed under standard laboratory conditions (22 °C–24 °C, 12 h light/dark cycle) with unrestricted access to food and water. For anesthesia, mice received an intraperitoneal injection of ketamine (Yuhan Ketamine 50 Inj., Yuhan Corporation, Korea) at 100 mg/kg, combined with xylazine (Rompun®, Bayer, Germany) at 10 mg/kg. In experiments requiring inhalation anesthesia, isoflurane (Hana Pharmaceutical Co., Korea) was administered at 2–5% in oxygen.

## Intrathecal Injection

IT injections were performed at the L4–L6 spinal level. A 30 G needle was inserted through the muscles to an appropriate depth, and the tail flick response was monitored to confirm needle placement. ICG (Tokyo Chemical Industry Co., Japan) stock solution was first prepared in dimethyl sulfoxide (DMSO) and then diluted in phosphate-buffered saline (PBS) to a final concentration of 1 mM for injection. ICG was administered using a syringe pump (Harvard Apparatus, US) at an infusion rate of 800 nL/min, with a total volume of 8 µL. For real-time imaging, the needle remained in place during the injection process. For all other imaging sessions, the needle was removed after the injection was completed. All IT injections and imaging procedures were conducted under 2∼5% ISO anesthesia with 500 mL/min oxygen inhalation. To investigate the effects of different anesthetic agents, a set of experiments was conducted in which IT injections were performed under K/X anesthesia.

## Intraparenchymal injection

To administer ICG into the brain parenchyma, mice were anesthetized with ISO and stereotaxic surgery was performed. A total of 2 µL of 1 mM ICG was injected into the striatum (AP + 0.5; ML −1.8; DV −3.6) using a Hamilton syringe connected to a stereotaxic syringe pump (Stoelting Co., US) at a controlled rate of 1 µL/min. After the injection, to prevent reflux the needle was kept in place for an additional 5 min before being slowly withdrawn.

## CM injection and intradermal forefoot injection

For CM injection, 8 µL of 1 mM ICG was administered to deliver the tracer into the CSF space, followed by PACT imaging (Supplementary Fig. 1). For sentinel lymph node (SLN) visualization, 200 µL of 1 mM ICG was intradermally injected into the mouse forefoot, and sagittal plane imaging was performed using PACT.

## CSF dynamics monitoring

To monitor CSF flow dynamics, we inserted a needle into the spinal canal of an ICR mouse, then secured it in the dorsal position on the PACT platform. Simultaneously with the start of the ICG infusion, we initiated single-volume PACT with 800 nm wavelength illumination at a 5 Hz frame rate. Each single volume captured a 3D image within a $12 \times 12 \times 12$ mm$^3$ FOV in isotropic resolution[20]. To compare the effects of K/X and ISO on glymphatic dynamics, PACT images were acquired

for 30 min after the infusion started. K/X was administered via IP injection, while ISO was delivered at a concentration of 2∼5% with an oxygen flow of 500 mL/min. To quantify the changes in PA signals within the CM, we manually designated polygonal ROIs for the CM on the XY-MAP images. The signals within the ROI were then sorted in descending order, and the average of the top 10–30% of the values was calculated to minimize the impact of overshooting and noise within ROIs[31,63]. Since dynamic PACT was performed using a single wavelength, we extracted the ICG signal by calculating the difference from the first frame within the CM regions. To improve the accuracy of this process, we used MATLAB's *imregister* function to register each frame's MAP image to the first frame's MAP image before extracting the signal. We also used the *imregister* function to generate a video of the CSF efflux to minimize the motion artifacts.

## IVIS Imaging

To validate the observations from PACT imaging, IVIS FL imaging was performed under identical experimental conditions. In vivo and ex vivo FL images of the whole body and brain were acquired using the IVIS imaging system (PerkinElmer, USA). All image acquisition and data analysis were performed using Living Image software (PerkinElmer, USA).

## Statistical Analysis

All PACT results were collected from experiments conducted on eight mice (three mice for fluorescence imaging). For ISO group mouse 4 in Fig. 2b (see Source Data file), missing data points were linearly interpolated for plotting the figure and analyzing data. Due to the loss of one AD mouse prior to the 96-hour monitoring, the sample size for the AD group at 96 h was reduced to 7. All statistical data are presented as mean ± standard errors. To illustrate the distribution of the datasets, individual animal data were overlaid as dots on the bar graphs. We used a two-tailed t-test to examine the statistical significance between experimental groups (paired test for Fig. 1f, h and independent test for Figs. 2b, 2c, 3c, 3d) with significance levels indicated as $*p < 0.05$, $**p < 0.01$, and $***p < 0.005$. Effect sizes were calculated using Cohen's $d$, defined as the mean difference divided by the standard deviation of the differences."

## Reporting summary

Further information on research design is available in the Nature Portfolio Reporting Summary linked to this article.

# Data availability

The main data supporting the findings of this study are included in the main text, figures, and supplementary information. Source data are provided with this paper.

# Code availability

All post-processing and quantifications were performed using custom MATLAB-based code. The MATLAB code used for blind spectral unmixing and ROI-based quantification is available at https://github.com/SeongwookChoi/PACT_CSF, with a permanent archived version at Zenodo (DOI: 10.5281/zenodo.18228433).

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

## Acknowledgements

We sincerely thank Eun-Ju Park for her assistance with the experiments using K/X anesthesia. S.C., J.K., and H.J. contributed equally to this work. The work was supported by grants from the National Research Foundation of Korea (2023R1A2C3004880, 2020R1A6A1A03047902, 2021M3C1C3097624, and RS-2022-NR070031), and by a Korea Medical Device Development Fund grant from the Korean government (the Ministry of Science and ICT; the Ministry of Trade, Industry and Energy; the Ministry of Health & Welfare; and the Ministry of Food and Drug Safety) (Project Number: 1711195277, RS-2020-KD000008). This work was also supported by a grant of the KHIDI, funded by the Ministry of Health & Welfare, Republic of Korea (RS-2023-00262321), and supported by an Institute of Information & Communications Technology Planning & Evaluation (IITP) grant funded by the Korea government(MSIT) (No.2019-0-01906, Artificial Intelligence Graduate School Program (POSTECH)) and a Korean Evaluation Institute of Industrial Technology (KEIT) grant funded by the Korean government (MOTIE), and by the BK21 FOUR program and the Glocal University 30 Project. H.J. is supported by the Hyundai Motor Chung Mong-Koo Foundation.

## Author contributions

S.C., J.K., H.J., C.K., and Y.C. contributed to the conceptualization of the study. S.C., J.K., and H.J. performed the overall experiments and data analysis. Y.L., G.L., W-M.J., and Y.C. conducted the IVIS FL imaging to validate PACT results. K.K., D.S.L., Y-S.K., J.S.L., and G.J.C. supported the preparation and supply of experimental materials. The project was supervised by C.K. and Y.C. All authors were involved in discussing the results and writing the manuscript.

## Competing interests

C.K. has financial interests in OPTICHO, which, however, did not support this work. All the other authors declare no other competing interests.
