## [Transparent Peer Review file · Nature Communications]

Photoacoustic computed tomography monitors cerebrospinal fluid dynamics and glymphatic function

Corresponding Author: Professor Chulhong Kim

Version 0:

Reviewer comments:

Reviewer #1

(Remarks to the Author)

The authors employ the PACT to dynamic monitor cerebrospinal fluid flow throughout the bodies of mice via injection of ICG. The study successfully evaluated the effects of anesthesia and Alzheimer's disease on cerebrospinal fluid circulation in mice. This method will advance the study of cerebrospinal fluid and lymphatic system circulation, potentially providing new way for neurodegenerative disease research.

1. The authors should include previous multi-wavelength photoacoustic microscopy employed to study the dynamic processes of cerebrospinal fluid in meningeal lymphatic vessels. The authors should discuss the unique features of the proposed method comparing with others.
2. In Figure 1, the labels (d) and (e) are used twice. Should "Fig. 1. PACT monitoring of brain CSF dynamics" be labeled as "Fig. 2"? The author should carefully proofread the manuscript.
3. Why is the photoacoustic amplitude comparison in the lower left corner of Figure 1 not presented as mean \pm standard error? All others are presented as mean \pm standard error.
4. In Figure 3, the ICG signal in AD mice initially decreased and then increased. Could the authors elucidate the underlying mechanisms or provide relevant explanations for this observation?
5. In the "CSF Dynamics Monitoring" section, why was an 800 nm laser chosen instead of a 796 nm laser? Additionally, what are the advantages of using 796 nm over 800 nm for blind unmixing?

Reviewer #2

(Remarks to the Author)

Comments for the authors:

The study reported by Choi et al. presents a promising application of photoacoustic computed tomography (PACT) for monitoring cerebrospinal fluid (CSF) dynamics and glymphatic function, which holds significant relevance for neurodegenerative diseases. The use of non-invasive PACT imaging to track indocyanine green (ICG) for whole-body 3D visualization of CSF pathways and real-time monitoring of CSF efflux is a notable strength, offering a valuable tool for advancing our understanding of brain waste clearance mechanisms.

This study also evaluated CSF drainage under different anesthetic conditions, and its application in wild-type versus Alzheimer's disease (AD) mouse models further enhance its translational potential. Given the established link between impaired CSF clearance and neurodegeneration, this work could have meaningful clinical applications, making it an alternative to conventional medical imaging like MRI or PET.

However, while this study demonstrates PACT's ability to visualize and quantify CSF dynamics, it would be beneficial to include more mechanistic data (e.g., how damage in lymphatic vessel integrity or changes in CSF pressure may influence drainage efficiency of glymphatic system). The authors should also compare PACT findings with established methods (e.g., two-photon microscopy or MRI-based CSF tracing) to further validate their results, since in-vivo fluorescence imaging by IVIS system is not considered a conventional strategy for measuring glymphatic dynamics. In addition, while murine models

are useful, the authors did not address potential challenges in translating PACT-based CSF imaging to clinical settings, which should be discussed further.

Additional suggestions to the authors:

Main text:

1. The pages of the main text should be numbered with line numbers.

Main section (Introduction section):

2. It is incorrect for the authors to claim that this is the first study to demonstrate PACT as a novel murine glymphatic/lymphatic imaging technique using ICG to monitor CSF dynamics and glymphatic function. A similar report was published in 2024 by Yang et al. (PMID: 38664374). It is surprising that the authors neglected this report and failed to mention it in the manuscript. In addition, in Yang et al, they showed successful detection of ICG in the meningeal lymphatic vessels, which was instead claimed by the authors as undetectable by PACT. This contradiction needs to be explained and discussed by the authors.

3. The statement: "First, we verify that whole-body PACT can precisely locate and identify the anatomy of the glymphatic/lymphatic system, providing functional insight into CSF drainage pathways" is an overstatement. While the use of PACT to show whole-body scale images is appreciated, the results fail to provide a systematic view of the glymphatic/lymphatic flowing system at a whole-body scale, being limited to only two regions: the spinal cord and cisterna magna of the mice.

4. The claim: "Second, by comparing the real-time glymphatic CSF effluxes of mice anesthetized with ketamine/xylazine (K/X) and those anesthetized with isoflurane (ISO), we confirm that PACT's real-time imaging capability enables the analysis of CSF flow dynamics in brain regions such as the transverse sinus and CM" needs qualification. Although PACT was used, the image resolution and findings provided in this study do not significantly advance the current understanding of glymphatic/lymphatic routes.

5. The statement: "Finally, we employ whole-body PACT to evaluate long-term CSF drainage by confirming impaired brain waste clearance in Alzheimer's disease (AD) transgenic mice" should be reconsidered. While an AD mouse model was used, the advantage of whole-body PACT was not clearly demonstrated, and the results largely replicate already known phenomena.

6. The claim: "By providing a novel and efficient method with high spatiotemporal resolutions for studying CSF dynamics in the glymphatic/lymphatic system" is problematic because the high spatiotemporal resolution feature was not adequately demonstrated.

Results section:

7. Figure 1 contains duplicated labels of panels (d and e). Such editing errors are unacceptable.

8. In Figure 1, panel e shows obviously non-specific signals on the left side of the intestine, a region that should be signal-free. This raises concerns about signal specificity in other regions indicated in this panel. The authors need to address this concern and explain how signal specificity is determined.

9. In Figure 1, panel g shows no visible signal in the CM area at both time points, yet the results still show numerical differences. This should be corrected by providing different sets of representative images to allow visual identification of signals in SC and CM areas.

10. To better demonstrate the glymphatic/lymphatic flow, CM injection should be performed followed by scanning the brains and meninges to reveal the dynamics of tracer flow.

11. The statement: "because K/X mimics sleep-like conditions" is inappropriate. The authors should clarify what they mean by "sleep-like conditions" with K/X treatment? Moreover, they should address whether ISO-anesthetized mice were not in similar conditions?

12. Figure 2 contains incorrect labeling. This error is unacceptable.

13. Figure 2, panel c needs to include representative images.

14. The following description should be moved to the Introduction section rather than appearing in Results: "Because the brain lacks a general lymphatic system, waste is cleared via CSF circulation in the glymphatic system, a specialized perivascular network that promotes the exchange of CSF and ISF, facilitating the removal of metabolic waste and neurotoxic proteins. Passing through the subarachnoid space, CSF flows into the deep brain, penetrating the brain parenchyma and mixing with the ISF. The combined CSF/ISF fluid clears waste and drains into the subarachnoid granulations or meningeal lymphatic vessels."

15. Data in all figures should be presented as bar plots with standard errors.

Discussion section:

16. The claim: "First, our PACT system directly visualized the pathways of CSF circulation within the subarachnoid space and observed CSF drainage after IT injection of ICG." is not fully supported. The study did not demonstrate "pathway" since the timepoint results only show similar changes at different analyzed locations. To claim pathways, sequential events should be clearly demonstrated by the data.

17. The statement: "however, it was unable to detect the meningeal lymphatic vessels and meningeal lymphatic nodes (e.g., deep cLNs, cLNs, iliac lymph nodes, and sLNs), which are the other routes for waste disposal" directly contradicts the findings of PMID: 38664374. Please address this issue.

Methods section:

18. Under "Animals", the authors need to provide the Jackson lab mouse strain number and institutional IACUC approval number.

19. Information about sex and age of animals need to be provided for all experiments.

20. Essential experimental details for reproducibility are missing and should be provided: needle size for intrathecal

injection, ICG concentration and vehicle used. The company and concentration of anesthetic (K/X) used in this study. 21. In the intraparenchymal injection paragraph, the authors mention using a controlled rate of 1 $\mu\text{L}/\text{min}$, which is unusually fast and can cause local tissue damage and affect glymphatic flow. There is also inconsistency between time mentioned in the main text (10 min) and in the Methods section (7 min). This needs clarification.

Statistical analysis:

22. This is a major concern. All experiments in this study appear to have been conducted as single experiment with only 3-4 mice per group. This limited sample size likely explains why the authors need to use one-tailed t-test instead of the more rigorous two-tailed t-test, which are standard for biological experiments. The authors need to increase animal numbers and demonstrate experimental reproducibility before PACT can be considered a reliable tool for glymphatic/lymphatic studies.

Reviewer #3

(Remarks to the Author)

In this work Choi et al., introduces photoacoustic computed tomography to monitor CSF fluid dynamics and glymphatic activity in the mouse. They test their imaging method cross different experimental cohorts: KX versus isoflurane anesthesia and WT versus AD pathology. They implement different tracer administration techniques i.e., intrathecal injection of the ICG as well as intraparenchymal (striatal) injection. A main message of their work is that the photoacoustic CT (PACT) imaging approach is superior to other imaging techniques in part because it is faster (increased temporal resolution).

Major results:

Fig. 1: Whole body PACT MAP images are presented of the mouse at 30min and 24h after intrathecal injection of ICG. Several body organs including rib and areas associated with the cisterna magna and spinal cord can be identified. The ICG signal associated with spinal cord and cisterna magna is reduced at 24 h compared to 30min. In vivo fluorescence imaging was used confirm signal at what is presumably the spinal cord.

Fig. 2 (labeled erroneously as Fig. 1): In vivo brain PACT images acquired before and at various time after IT injection of ICG in mice anesthetized with KX or Isoflurane. CSF signal is measured at the level of the cisterna magna and shows that the signal is higher in KX anesthetized mice compared to isoflurane.

Fig. 3: An intraparenchymal injection is made into the striatum of normal and AD mice and the tracer mass is followed by PACT imaging over 96h. The disappearance of the signal is slower in the AD compared to WT mice.

Comments:

This is potentially interesting work. Below are comments and suggested improvements to the work of the group.

- Information related to the spatial resolution of the PACT images is lacking. This point is impossible to interpret based on description in the method section. The statement: "Each single volume captured a 3D image within a $12 \times 12 \times 12 \text{ mm}^3$ FOV in isotropic resolution" is not informative.
- Information related to image processing is also lacking. There is clearly gross motion in the images (videos). How did the authors correct for motion artefacts? The authors mention the MATLAB's imregister function to register each frame image – if this is used as motion correction they should state it clearly.
- The authors state: "The signals within the ROI were then sorted in descending order, and the average of the top 10-30% of the values was calculated." While this might be appropriate it will be necessary for the authors to present a sensitivity analysis to determine if this threshold is meaningful. Also, why the range? i.e., why not a cut-off? Please explain.
- The statistical analysis and sample size is problematic. The effect size of the KX vs isoflurane has been documented to be large in several other studies so this may be reflected in the data showing difference. However, it is not rigorous (and in fact misleading) to use a one-tailed t-test. Also, even with SEM the error bars are large for the AD group at 24 hrs. Please show images of all the 3 time points of the AD mice (at least for 2 of the 3) so that the reviewer can evaluate the robustness of the technique. This data are not convincing given the very limited sample size.
- There is no mention of physiological monitoring during imaging. Can the authors please elaborate on this matter especially body temperature.
- Why do the authors talk about drainage pathways when they do not seem to measure these in the current study. This should be toned down.
- The PACT technique has high(er) temporal resolution – what is the temporal resolution? And how did this feature improve the glymphatic function measurements... the authors should comment on this since they emphasize this point.

Reviewer #4

(Remarks to the Author)

NCOMMS-25-18953 Photoacoustic computed tomography monitors cerebrospinal fluid dynamics and glymphatic function Choi, et al.

Summary:

In this communication, authors present PACT images along with validating fluorescence images of ICG following intrathecal administration in WT and FAD mice as well as isoflurane and ketamine/xylazine-anesthetized mice to visualize differences in CSF dynamics and glymphatic function. To assess CSF dynamics, PACT signals are unmixed to segregate out acoustic signals due to Hb to provide signals associated with ICG. The results were very interesting, highlight the current limitations of PACT for assessing lymphatic function. At this stage, PACT does not provide any new insights into CSF dynamics, but the communication describes capabilities. The communication on the use of PACT is important, especially since PACT has been performed in the lower extremity lymphatics in humans, and as such, the contribution provides a potentially exciting

approach to visualize extracranial CSF outflow. The following comments are made to enhance the impact of the contribution:

Major Comments:

1. The physiology of CSF drainage is not correctly described in the contribution. The authors perform an intrathecal injection which is a retrograde introduction of contrast. While intrathecal is preferred over CM injections (because of a smaller disturbance of subarachnoid pressures), it nonetheless does not represent physiological CSF flow. Cranial CSF dynamics represent flow from the ventricular spaces into the subarachnoid spaces and into the meningeal lymphatics for collection through the deep jugular lymphatics and lymph nodes for ultimate clearance through the hepatobiliary system. CSF dynamics in the spinal canal show efflux through lymphatics into the renal lymph nodes. Rodents and infants do not have arachnoid granulations, which have been historically associated with extracranial outflow through the venous sinuses in humans and larger mammals. The contribution needs to be amended with reference to transverse sinuses.
2. After intrathecal ICG injections of μM (not mM as in this study), the jugular lymph nodes and drainage through the olfactory bulb are clearly visible via fluorescence imaging almost immediately after injection (see cited references). In the PACT images shown, no lymph nodes or lymphatic vessels were visualized and more surprisingly, the liver does not show ICG PACT signals – suggesting the unmixing process is severely hampered. Is the conducting lymphatic vessel and the axillary and inguinal LNs visualized on the medial side after hind dorsal foot ICG intradermal injection?
3. I am wondering if the authors looked more carefully at the liver, they might see early ICG PACT signal in K/X anesthetized and WT mice compared to isoflurane anesthetized and FAD mice. I note that the venous sinus tends have an increased PCT signal 1 hour after ICG administration, which is very consistent with extracranial drainage from the lymphatics and into the hemovascular system. Because mice lack arachnoid granulations, it is unlikely that this is uptake directly into the venous system.
4. I would argue that the PACTs visualization of the CM and spinal cord alone is woefully inadequate to assess CSF dynamics in the brain or its extracranial outflow (see for example PMID 374403980). The statement that “PACT’s real-time imaging capability enables the analysis of CSF flow dynamics in brain regions such as the transverse sinus and CM” may be inferior to fluorescence and PET techniques. While others have visualized reduced efflux through the cervical LNs in FAD mice and under isoflurane, this contribution uses the elevated PACT signal in the CM as surrogate for reduced extracranial clearance. This should be clearly stated in the discussion within a revised manuscript. On the other hand, the intraparenchymal injection of ICG does indeed directly correspond to studies of AD and anesthesia found in the literature and is exciting to see.
5. While retention of ICG in the CM after intrathecal injection is interesting, it really does not reflect impaired extracranial outflow as would be seen from the ICG signal in the jugular lymph nodes.
6. The authors perform dorsal imaging, but wouldn’t ventral imaging provide better signals to visualize olfactory bulb and cervical lymph node drainage? I would think that the ventral imaging would be more interesting than the dorsal imaging.
7. “First our PACT system directly visualized the pathways of CSF circulation within the subarachnoid space and observed CSF drainage after IT injection of ICG” ; PACT could image the clearance of waste...”-- This is not supported by the results. PACT system enabled detection of changing ICG concentrations in CM and spinal canal only after intrathecal injection and changes in parenchymal concentrations upon intraparenchymal injection.
8. ICG remains in lymph nodes in humans and mice and can be visualized using FLI. It may be that the sensitivity of the unmixed PACT is not sufficient to detect LNs? Likewise, the subarachnoid space was not well contrasted in PACT as it is in FLI. Perhaps this is a sensitivity issue for PACT?
9. Finally, after reviewing the results, the introduction describing FLI seems incorrect: “FLI lacks depth information and the spatial resolution of in vivo imaging, restricting 3-D analysis for accurate tracer monitoring.” As stated above FLI in references 10-12 do show lymph nodes, lymphatic vessels and jugular lymph nodes have been clinically seen in adults after ICG dosage that is 100 fold less than used herein. So, since PACT does not show these structures, it might be best to omit such statements.
10. The authors are congratulated for their pioneering communication.

Minor comments:

1. While the authors are commended for using SI units for their IVUS fluorescence measurements, the proper measurements for excitation irradiance is W/cm^2 and for fluorescence irradiance is $\text{W}/\text{cm}^2/\text{sr}$.
2. How is the coupling of the transducer performed in these studies?
3. In the text, the pink and blue arrows are switched. It is correct in the figure caption.
4. Results, 4th line, “brain adipose tissue”? Can you please explain this? This looks like BAT to me.
5. Absorption characteristics of ICG in the CSF (which does not have a high plasma protein concentration) should not be the reasons for unmixing difficulties. Perhaps to test, the use of Evan’s Blue dye can be used? However, Evan’s Blue dye at

high concentrations will show neurotoxicity.

6. Can you tell me what the resolution of the PACT system is, as used in the communication?

Eva Sevick-Muraca

Version 1:

Reviewer comments:

Reviewer #1

(Remarks to the Author)

The authors have addressed most of my comments. It can be considered to be accepted.

Reviewer #2

(Remarks to the Author)

The authors have adequately addressed all my previous concerns.

As a final suggestion, I would recommend that all figures be presented as dot plots rather than bar graphs, in order to better illustrate the distribution of the datasets.

Reviewer #3

(Remarks to the Author)

Reviewer #5

(Remarks to the Author)

While the authors gave point-by-point responses to the major comments of reviewer 4, it appears that some major points were not addressed. Below are my concerns, that were included and integrated into multiple points from Reviewer 4:

(1) The authors do not acknowledge that rodents do not have arachnoid granulations, and the authors do not seem to specify other CSF efflux routes from the skull including peri-neuronal or nasal efflux.

(2) The biggest caveat of this manuscript is that measurements of ICG in the CM may not be glymphatic influx but an artifact of isoflurane-induced swelling of the brain and spinal cord, limiting intrathecal distribution of tracer throughout the subarachnoid space under isoflurane. Can the authors differentiate between the two hypotheses?

(3) I am confused about the argument that PAM is unsuitable because of the imaging window, but then the authors say their technique cannot do real-time imaging unless they make the imaging window small (confined to the brain)

(4) Additionally, the authors argue that the spectral unmixing is critical to get depth. If they take that out for the imaging, how is what they're doing more insightful than FLI? I am sure there is a rationale, but it is unclear.

(5) The authors state in their response to review that sensitivity is not an issue for PACT, yet also state that they cannot pick up signals that are clearly visible with FLI. The explanation given is not convincing about the superiority of PACT over FLI.

(6) The authors never address nor attempt ventral imaging as suggested by Reviewer 4.

Moderate-to-major concerns based on this reviewer's read-through:

(1) Figure 3 is probably the most informative and novel figure of the paper. The description of this figure in the results should be re-written for conciseness. Several of the paragraphs are redundant. The bar charts are detracting from the rigor of the figure. Throughout the manuscript, figures should have individual animal information. With such small sample sizes throughout there is no reason to have bar chart plots.

(2) The intro is four pages long, yet the discussion is only two pages with a large portion devoted to rehashing results. Is there a way to rewrite to shorten the introduction, minimize extensive rehashing of results/bring attention to how PACT has both important new innovations and significant caveats in the discussion?

(3) The references for the anesthesia effect (the authors use numbers 12 and 13) are incorrect, or, at minimum, not what the field cites for the effect of anesthesia on glymphatic function. More relevant citations should be included, if not used to replace these two.

(4) Reference 13 for the AD mouse model may not be the most applicable/suitable reference. Can the authors provide additional examples that are measuring perivascular flow and tracer movement within the brain instead of just the lymphatic system?

Minor concerns:

(1) May want to double-check use of ISF acronyms in intro

(2) The color of the arrows in Figure 1 may need to be more saturated.

Version 2:

Reviewer comments:

Reviewer #5

(Remarks to the Author)

This revised manuscript has successfully addressed concerns. Thank you for a fun paper!

Original manuscript title: “Photoacoustic computed tomography monitors cerebrospinal fluid dynamics and lymphatic function”

Authors and affiliations

Seongwook Choi ^{a, †}, Jiwoong Kim ^{a, †}, Hyeonsoo Jeon ^{a, †}, Yejin Lee ^{b, c}, Gyeongju Lim ^b, Won-Min Ju ^b, Kyuwan Kim ^d, Dong Soo Lee ^a, Yun-Sang Lee ^{b, c}, Jae Sung Lee ^{b, d}, Gi Jeong Cheon ^{b, c}, Yoori Choi ^{b, c, *}, and Chulhong Kim ^{a, *}

^a Department of Electrical Engineering, Convergence IT Engineering, Medical Science and Engineering, Institute of Artificial Intelligence, and Medical Device Innovation Center, Pohang University of Science and Technology, Republic of Korea

^b Department of Nuclear Medicine, Seoul National University Hospital, Seoul, Republic of Korea

^c Department of Molecular Medicine and Biopharmaceutical Sciences, Graduate School of Convergence Science and Technology, College of Medicine or College of Pharmacy, Seoul National University, Seoul, Republic of Korea

^d Department of Nuclear Medicine, College of Medicine, Seoul National University, Seoul, Republic of Korea

[†]These authors contributed equally to this work.

*Corresponding authors: chulhong@postech.edu, yms086@snu.ac.kr

Dear Editor,

Thank you for coordinating the review of our manuscript. We also thank the reviewers for their comprehensive and critical evaluations. In response to the reviewers' comments, **we have revised the manuscript and added new text, as highlighted in yellow in the accompanying version**. Our responses to the reviewers' specific comments are also below, where the reviewers' comments are in normal font, our responses are in italics, and revised text is quoted in italics.

Sincerely,

The Authors

Reviewer 1

General comments: The authors employ the PACT to dynamic monitor cerebrospinal fluid flow throughout the bodies of mice via injection of ICG. The study successfully evaluated the effects of anesthesia and Alzheimer's disease on cerebrospinal fluid circulation in mice. This method will advance the study of cerebrospinal fluid and lymphatic system circulation, potentially providing new way for neurodegenerative disease research.

Reply: *Thank you for positive comments.*

Specific comments:

1. The authors should include previous multi-wavelength photoacoustic microscopy employed to study the dynamic processes of cerebrospinal fluid in meningeal lymphatic vessels. The authors should discuss the unique features of the proposed method comparing with others.

Reply:

We appreciate your constructive comments. As you mentioned, previous studies have employed multi-wavelength photoacoustic microscopy (PAM) to explore the dynamic processes of cerebrospinal fluid (CSF) movement in meningeal lymphatic vessels [Refs. 1-2]. These studies have made significant contributions to our understanding of the localization and movement of CSF. However, despite the strengths of PAM and the novel findings it has provided, PAM does have inherent limitations.

First, while PAM achieves high-resolution, micro-scale imaging, it primarily provides non-real-time imaging data. This restricts the ability to dynamically monitor and quantify CSF flow under various physiological conditions, making it difficult to capture rapid physiological changes in homeostasis.

Second, previous PAM studies have predominantly focused on localized brain imaging. This focus has restricted the understanding of systemic physiological interactions, and it does not provide a comprehensive view of CSF distribution in major lymphatic pathways on a macro scale.

In contrast, our study has some unique aspects. Our PACT system enables 3D real-time monitoring within the brain-size region of interest (ROI), allowing dynamic, in vivo visualization and quantification of CSF flow under various conditions, which is crucial for understanding rapid physiological changes in homeostasis.

Furthermore, our research employs whole-body PACT following indocyanine green (ICG) injection to visualize and evaluate CSF distribution in major lymphatic pathways on a macro scale. This comprehensive approach offers valuable insights into systemic physiological interactions and their implications for neurodegenerative diseases, thus providing a broader understanding beyond localized brain imaging.

For the first time, our research visualizes whole-body 3D CSF distribution in major lymphatic pathways using PACT following intrathecal ICG injection. This innovative approach enables a comprehensive mapping of CSF efflux, which is useful for correlating glymphatic function with the progression of neurodegenerative diseases.

Additionally, using real-time PACT, we dynamically monitor variations in CSF flow under different anesthetic conditions. These studies provide foundational knowledge on how anesthetics influence CSF dynamics, which is crucial for clinical applications and understanding glymphatic function under various states.

[Ref. 1] Wang, Z. et al., Monitoring the perivascular cerebrospinal fluid dynamics of the glymphatic pathway using co-localized photoacoustic microscopy. Optics Letters 48.9, 2265-2268 (2023).

[Ref. 2] Yang, F. et al., Advancing insights into in vivo meningeal lymphatic vessels with stereoscopic wide-field photoacoustic microscopy. Light: Science & Applications 13.1, 96 (2024).

To summarize the above points, we added the following sentences to the Introduction, on Page 4, Line 101:

“Meanwhile, photoacoustic microscopy (PAM) is another imaging technique that utilizes the PA effect, similar to PACT, but focuses on micro-scale imaging as opposed to PACT’s macro-scale imaging. There have been studies employing PAM to explore multi-wavelength imaging of the dynamic processes of CSF in meningeal lymphatic vessels. These studies have contributed valuable data on the localization and movement of CSF. However, they have several limitations. First, previous PAM studies primarily provided non-real-time imaging data, which limits the ability to dynamically monitor and quantify CSF flow under various physiological

conditions. This limitation makes it difficult to capture rapid physiological changes in homeostasis. In other words, these studies lack the capability to dynamically monitor variations in CSF flow, which is crucial for understanding the glymphatic function. Second, previous research has predominantly focused on localized brain imaging, which restricts the understanding of systemic physiological interactions and cannot provide a comprehensive view of CSF distribution in major lymphatics on a macro scale. Consequently, while previous PAM studies have laid important groundwork for CSF imaging using the PA effect, there remains an opportunity to develop more holistic, real-time, and comprehensive insights into CSF dynamics and glymphatic function via PACT.”

2. In Figure 1, the labels (d) and (e) are used twice. Should "Fig. 1. PACT monitoring of brain CSF dynamics" be labeled as "Fig. 2"? The author should carefully proofread the manuscript.

Reply: Thank you for catching this. We have corrected the error and proofread the whole manuscript again.

3. Why is the photoacoustic amplitude comparison in the lower left corner of Figure 1 not presented as mean \pm standard error? All others are presented as mean \pm standard error.

Reply: In the original manuscript, this experiment was conducted using a single mouse. However, in response to the valuable comments from the reviewers, we have repeated the experiment with a statistically sufficient sample size ($N = 8$) and performed quantitative analysis accordingly. The corresponding figure has been updated to display the data as mean \pm error. Please refer to the response to Reviewer #2's Comment 22 for further details.

4. In Figure 3, the ICG signal in AD mice initially decreased and then increased. Could the authors elucidate the underlying mechanisms or provide relevant explanations for this observation.

Reply: To ensure more rigorous statistical validation, we conducted additional experiments, increasing the sample size to $N=8$. As a result, a slight change in the trend was observed. Please refer to the response to Reviewer #2's Comment 22 for further details.

5. In the "CSF Dynamics Monitoring" section, why was an 800 nm laser chosen instead of a 796 nm laser? Additionally, what are the advantages of using 796 nm over 800 nm for blind unmixing ?

Reply: For dynamic imaging, we utilized an 800-nm wavelength, which is widely adopted for structural photoacoustic imaging. This wavelength also provides a strong signal for indocyanine green (ICG), making it suitable for capturing real-time dynamics beyond wavelength-specific constraints. Furthermore, our previous studies successfully demonstrated both structural and ICG-enhanced PACT imaging using an 800 nm wavelength, and they effectively extracted ICG signals by differentiating PACT signals [Ref. 1-2].

For spectral unmixing, we selected 796 nm, which corresponds to the near isosbestic point of oxyhemoglobin and deoxyhemoglobin, consistent with its widespread use in prior preclinical and clinical studies [Ref. 3-5]. In the context of blind unmixing techniques, the emphasis shifts from individual wavelengths to the entire absorption spectrum of each chromophore, as extracted from multi-wavelength imaging data. Therefore, from a theoretical perspective, the specific choice of wavelength is not critical, since the unmixing process leverages the full spectral information to distinguish between different absorbers.

[Ref. 1] Choi, S. et al. Deep learning enhances multiparametric dynamic volumetric photoacoustic computed tomography in vivo (DL-PACT). *Advanced Science* 10, 2202089 (2023).

[Ref. 2] Yang, J. et al. Multiplane Spectroscopic Whole-Body Photoacoustic Computed Tomography of Small Animals In Vivo. *Laser & Photonics Reviews* n/a, 2400672, doi:<https://doi.org/10.1002/lpor.202400672>.

[Ref. 3] Kim, J. et al. 3d multiparametric photoacoustic computed tomography of primary and metastatic tumors in living mice. *ACS nano* 18, 18176-18190 (2024).

[Ref. 4] Kim, J. et al. Multiparametric photoacoustic analysis of human thyroid cancers in vivo. *Cancer Research* 81.18, 4849-4860 (2021).

[Ref. 5] Park, B. et al. 3D wide-field multispectral photoacoustic imaging of human melanomas in vivo: a pilot study. Journal of the European Academy of Dermatology and Venereology 35.3, 669-676 (2021).

Reviewer 2

General comments: The study reported by Choi et al. presents a promising application of photoacoustic computed tomography (PACT) for monitoring cerebrospinal fluid (CSF) dynamics and glymphatic function, which holds significant relevance for neurodegenerative diseases. The use of non-invasive PACT imaging to track indocyanine green (ICG) for whole-body 3D visualization of CSF pathways and real-time monitoring of CSF efflux is a notable strength, offering a valuable tool for advancing our understanding of brain waste clearance mechanisms.

This study also evaluated CSF drainage under different anesthetic conditions, and its application in wild-type versus Alzheimer's disease (AD) mouse models further enhance its translational potential. Given the established link between impaired CSF clearance and neurodegeneration, this work could have meaningful clinical applications, making it an alternative to conventional medical imaging like MRI or PET.

However, while this study demonstrates PACT's ability to visualize and quantify CSF dynamics, it would be beneficial to include more mechanistic data (e.g., how damage in lymphatic vessel integrity or changes in CSF pressure may influence drainage efficiency of glymphatic system). The authors should also compare PACT findings with established methods (e.g., two-photon microscopy or MRI-based CSF tracing) to further validate their results, since in-vivo fluorescence imaging by IVIS system is not considered a conventional strategy for measuring glymphatic dynamics. In addition, while murine models are useful, the authors **did not address potential challenges** in translating PACT-based CSF imaging to clinical settings, which should be discussed further.

Reply: Thank you for your insightful comments. We appreciate your suggestions and would like to address them as follows:

We acknowledge that the inclusion of more mechanistic data would augment our study. However, our research serves as an important foundational step in demonstrating the potential of PACT for monitoring CSF dynamics within the glymphatic and lymphatic systems of the entire body. Our findings highlight the capabilities of PACT, and we anticipate that this work will stimulate further research incorporating new physiological and mechanistic insights, leveraging the high spatiotemporal resolution of the PACT modality. Please also see the response to your Comment 4. We also commented on the advantage of PACT's high spatiotemporal resolution in the responses to your Comment 6 and Reviewer #3's Comment 7.

PACT has already shown promise in clinical settings. Notably, PACT has been successfully performed on the lower extremity lymphatics in humans [Ref. 1], demonstrating its potential for imaging CSF pathways. As PACT is based on ultrasound imaging, it offers advantages for clinical translation, including ease of implementation and safety. Recent advances have led to PACT's widespread use in clinical trials, and its approval by the FDA for breast cancer imaging further underscores its integration into mainstream medical practice [Ref. 2]. These developments suggest that PACT is increasingly becoming part of regulated healthcare protocols, highlighting its potential for CSF monitoring in clinical settings. However, PACT's penetration depth is inherently limited, which restricts its ability to visualize deeper lymphatic structures, such as deep lymph nodes and vessels. For brain imaging in particular, the skull poses additional challenges of acoustic attenuation and distortion, limiting the achievable imaging depth and quality. Recent advances, including transcranial PACT demonstrations in humans by Wang et al. [Ref. 3], indicate the potential for future clinical translation of PACT in brain and CSF applications. To overcome these clinical limitations, contrast agents for PACT are actively being developed to enhance imaging sensitivity and extend its applicability to deeper tissues [Ref. 4].

We added this opinion about clinical settings in the Discussion Section:

Page 16, Line 326:

“Furthermore, for clinical translation, PACT's demonstrated efficacy in clinical settings, including its successful use for imaging human lymphatics and the brain imaging, and its obtaining FDA approval, shows its potential for human CSF monitoring.”

Page 17, Line 337:

“Additionally, these advancements suggest that PACT could offer valuable insights into CSF drainage in clinical contexts, taking advantage of better sensitivity for deeper tissue imaging.”

[Ref. 1] Suzuki, Y. et al. Subcutaneous lymphatic vessels in the lower extremities: comparison between

photoacoustic lymphangiography and near-infrared fluorescence lymphangiography. Radiology 295.2, 469-474 (2020).

[Ref. 2] Park, J. et al. Clinical translation of photoacoustic imaging. Nature Reviews Bioengineering, 1-20 (2024).

[Ref. 3] Na, S., Russin, J. J., Lin, L., Yuan, X., Hu, P., Jann, K. B., ... & Wang, L. V. Massively parallel functional photoacoustic computed tomography of the human brain. Nature Biomedical Engineering, 6(5), 584-592. (2022)

[Ref. 4] Choi, W., Park, B., Choi, S., Oh, D., Kim, J., & Kim, C. Recent advances in contrast-enhanced photoacoustic imaging: overcoming the physical and practical challenges. Chemical Reviews, 123(11), 7379-7419. (2023)

We would like to point out that fluorescence imaging (FL) is indeed a widely utilized method in many CSF lymphatic/glymphatic imaging studies, as evidenced by numerous references in the literature [Ref. 5-8]. FL is considered a conventional strategy for measuring glymphatic dynamics and has been extensively validated in various contexts. Furthermore, it is important to note that comparing MR tracers and PACT tracers presents a challenge due to the differences in their physiological impacts. Consequently, validating PACT results by using MRI results is not feasible. On the other hand, FL and PACT both utilize ICG, allowing for a more accurate comparison of imaging performance between these modalities. In summary, we argue that FL validation is widely recognized as a conventional method in this field, and that validation using MRI is not feasible due to uncontrollable tracer differences.

[Ref. 5] Kwon, S. et al. Fluorescence imaging of lymphatic outflow of cerebrospinal fluid in mice. Journal of Immunological Methods 449, 37-43 (2017).

[Ref. 6] Hablitz, L. M. et al. Increased glymphatic influx is correlated with high EEG delta power and low heart rate in mice under anesthesia. Science advances 5, eaav5447 (2019).

[Ref. 7] Shibata, Y. et al. Imaging of cerebrospinal fluid space and movement in mice using near infrared fluorescence. Journal of neuroscience methods 147.2, 82-87 (2005).

[Ref. 8] Shibata, Y. et al. Imaging of cerebrospinal fluid space and movement of hydrocephalus mice using near infrared fluorescence. Neurological sciences 28.2, 87-92 (2007).

Specific comments:

1. The pages of the main text should be numbered with line numbers.

Reply: *Thank you for your comment.*

2. It is incorrect for the authors to claim that this is the first study to demonstrate PACT as a novel murine glymphatic/lymphatic imaging technique using ICG to monitor CSF dynamics and glymphatic function. A similar report was published in 2024 by Yang et al. (PMID: 38664374). It is surprising that the authors neglected this report and failed to mention it in the manuscript. In addition, in Yang et al, they showed successful detection of ICG in the meningeal lymphatic vessels, which was instead claimed by the authors as undetectable by PACT. This contradiction needs to be explained and discussed by the authors.

Reply: *Thank you for these comments. Please see the response to Reviewer #1's Comment 1*

3. The statement: "First, we verify that whole-body PACT can precisely locate and identify the anatomy of the glymphatic/lymphatic system, providing functional insight into CSF drainage pathways" is an overstatement. While the use of PACT to show whole-body scale images is appreciated, the results fail to provide a systematic view of the glymphatic/lymphatic flowing system at a whole-body scale, being limited to only two regions: the spinal cord and cisterna magna of the mice.

Reply: *We appreciate the precision of your comment, and we have revised the sentence to more accurately reflect the scope of our results as follows, on Page 4, Line 117:*

“First, we demonstrate that whole-body PACT can effectively visualize CSF dynamics in key lymphatic regions, including the subarachnoid space adjacent to the spinal cord, as well as the CM. This capability highlights PACT’s potential to provide functional insights into CSF drainage pathways.”

4. The claim: “Second, by comparing the real-time glymphatic CSF effluxes of mice anesthetized with ketamine/xylazine (K/X) and those anesthetized with isoflurane (ISO), we confirm that PACT’s real-time imaging capability enables the analysis of CSF flow dynamics in brain regions such as the transverse sinus and CM” needs qualification. Although PACT was used, the image resolution and findings provided in this study do not significantly advance the current understanding of glymphatic/lymphatic routes.

Reply: *Thank you for constructive comments. First of all, we have modified a typo, changing ‘effluxes’ to ‘influxes’ on Page 5, Line 117:*

“Second, by comparing the real-time glymphatic CSF influxes of mice anesthetized with ketamine/xylazine (K/X) and those anesthetized with isoflurane (ISO), we confirm that PACT’s real-time imaging capability enables the analysis of CSF flow dynamics in brain regions such as the transverse sinus and CM.”

Meanwhile, we emphasize that our study focuses on establishing PACT’s capability to monitor CSF macro-scale dynamics within 3D real-time imaging in high spatiotemporal resolution. Although, as we mentioned in the Discussion Section, PACT suffers lower sensitivity to the tracer than MRI, PACT still has advantages such as using simpler and more biocompatible contrast agents like methylene blue (MB), indocyanine green (ICG), and Evans Blue, whereas MRI often requires more complex and potentially toxic contrast agents. Second, PACT offers high spatiotemporal resolution, enabling detailed, real-time imaging of dynamic physiological processes. In contrast, conventional MRI lacks real-time imaging capabilities, and its spatial resolution for small-animal applications remains constrained. Standard MRI systems operating at 3 T typically achieve spatial resolutions of approximately $1 \times 1 \times 1 \text{ mm}^3$, while high-field 7-T MRI systems can achieve sub-millimeter resolutions approaching $500 \mu\text{m}$ [Ref. 1]. However, these values still do not reach the PACT specification, and even advanced MRI configurations struggle to achieve the temporal resolution necessary for capturing rapid physiological dynamics, thereby limiting their utility for real-time monitoring of fast biological processes. Lastly, PACT systems are more cost-effective, compact, and portable than large and immobile MRI or PET machines, providing greater flexibility in different environments. (Please also see the response to your General Comment.)

[Ref. 1] Duyn, J. H. The future of ultra-high field MRI and fMRI for study of the human brain. Neuroimage, 62(2), 1241-1248. (2012).

On the other hand, while we acknowledge that our study may not present entirely novel physiological insights, it does offer technical achievements and potential clinical implications that advance the current understanding of glymphatic/lymphatic routes.

First, our research introduces a developed PACT system capable of real-time CSF imaging during intrathecal injection. This capability allows for dynamic observation of CSF flow, providing a more immediate and practical understanding of physiological changes than static imaging methods. This real-time monitoring can be particularly valuable in both research and clinical settings, where immediate feedback on CSF dynamics is crucial.

Second, we applied a blind unmixing technology to accurately extract and quantify the amount of ICG in the brain. This technique enhances the precision of our measurements and ensures that the data collected is reliable and applicable for disease-related studies. Accurate quantification is essential for understanding CSF dynamics and glymphatic function, especially in the context of neurodegenerative diseases such as Alzheimer’s disease.

Furthermore, the potential clinical implications of our study are significant. The ability to monitor CSF flow and glymphatic function in real-time can inform future therapeutic strategies and interventions. For example, understanding how different anesthetics or pathological conditions affect CSF dynamics could lead to improved anesthesia protocols or early diagnostic tools for neurodegenerative diseases.

While the sensitivity of PACT may not yet match that of other established techniques like MRI and PET, its real-time imaging capability, non-invasiveness, and use of simpler, more biocompatible contrast agents represent significant advancements. These improvements support more frequent and less invasive monitoring, which

could have substantial implications for both basic research and clinical practice. We hope these points address your concerns and highlight the unique technical contributions and potential clinical applications of our study to the field of glymphatic/lymphatic research. We modified the Introduction on Page, Line 78 to improve understanding:

“However, these conventional imaging methods have inherent limitations. FLI lacks depth information and the spatial resolution of in-vivo imaging, restricting 3-D analysis for accurate tracer monitoring. While MRI can provide 3-D imaging capabilities, standard clinical systems are limited by poor spatial resolution, and achieving high spatial resolution requires expensive premium high-field MRI systems, which still cannot provide real-time imaging due to inherently long acquisition times and high operational costs, making it impractical for dynamic CSF tracking. PET lacks detailed anatomical information and relies on radioactive isotopes, raising safety concerns. Additionally, PET requires multimodal approaches, such as X-ray computed tomography or MRI, to compensate for its limited anatomical resolution. The ideal imaging modality for tracking CSF would provide high resolution, a high frame rate, and comprehensive 3-D structural information.”

5. The statement: “Finally, we employ whole-body PACT to evaluate long-term CSF drainage by confirming impaired brain waste clearance in Alzheimer’s disease (AD) transgenic mice” should be reconsidered. While an AD mouse model was used, the advantage of whole-body PACT was not clearly demonstrated, and the results largely replicate already known phenomena.

Reply: Please see the response to Comment 4 about the advantage of whole-body PACT.

6. The claim: “By providing a novel and efficient method with high spatiotemporal resolutions for studying CSF dynamics in the glymphatic/lymphatic system” is problematic because the high spatiotemporal resolution feature was not adequately demonstrated.

Reply: Please see the response to Comment 4 and Reviewer #3’s Comment 7 about the advantage of PACT’s high spatiotemporal resolution.

In the Methods, we have detailed the spatial and temporal resolutions of our PACT system, referencing sources 16 and 27. Specifically, the spatial resolution is approximately 380 μm . The temporal resolution, determined by the laser repetition rate and the data acquisition board’s channel number, is 5 Hz. We have added the spatial resolution of the PACT system to the Methods section, on Page 18, Line 355:

“The developed PACT system effectively detects omnidirectionally propagating PA waves using a customized 1024-element hemispherical US transducer array (Japan Probe, Inc., Japan), and it achieves an isotropic spatial resolution of 380 μm .”

7. Figure 1 contains duplicated labels of panels (d and e). Such editing errors are unacceptable.

Reply: Thank you for your comment. We have corrected the error and proofread the whole manuscript again.

8. In Figure 1, panel e shows obviously non-specific signals on the left side of the intestine, a region that should be signal-free. This raises concerns about signal specificity in other regions indicated in this panel. The authors need to address this concern and explain how signal specificity is determined.

Reply: Thank you for the comment. This artifact arises from feces in the intestine that saturate the digitized PA signal. Consequently, the PA signal at each wavelength does not accurately represent the absorption spectrum of the material components. Such artifacts can be mitigated by fasting the mouse or by reducing the amplifier gain, but these measures could induce other artifacts, from the mouse’s physiological condition or the low SNR of the PA signal, respectively. For clarity, we have added the following sentences on Page 7, Line 166, and we also added a clarifying indication to Figure 1:

“Note that the ICG signal observed in the intestine in Figure 1(e) is influenced by artifacts. These artifacts arise from feces in the intestine (purple arrow) causing saturation of the digitized PA signal. As a result, for this kind

of strong absorbers, the PA signal at each wavelength does not accurately represent the absorption spectrum of the material components, and the artifact occurs.”

9. In Figure 1, panel g shows no visible signal in the CM area at both time points, yet the results still show numerical differences. This should be corrected by providing different sets of representative images to allow visual identification of signals in SC and CM areas.

Reply: We have added Supplementary Figure 3 to provide a zoomed fluorescence image of the CM and brain area with an adjusted color scale range to visualize CM's fluorescence signal's difference between the 30-minute and 24-hour time points:

on Page 8, Line 183:

“To more clearly visualize the FL signal in the CM region, Supplementary Figure 4 presents an enlarged view of the CM with an adjusted color scale. It was confirmed that the FL signals in the CM had completely disappeared at 24 hours in the images displayed using the same colormap.”

On Supplementary Information,

Supplementary Figure 4. Fluorescence image of the cisterna magna and brain region following intrathecal injection.

10. To better demonstrate the glymphatic/lymphatic flow, CM injection should be performed followed by scanning the brains and meninges to reveal the dynamics of tracer flow.

Reply: Thank you for your helpful comment. To further demonstrate the potential of PACT for CSF assessment, we have included additional experiments in Supplementary Figure 1. A volume of 8 μ L of 1 mM ICG, equivalent to that used for intrathecal injection, was directly administered into the CM. A strong increase in PA signal intensity in the CM due to ICG was observed 30 minutes post-injection. After 24 hours, ICG clearance from the CM via CSF drainage was confirmed. However, visualizing the lymph nodes or additional pathways remained difficult. This limitation was likely due to the sensitivity constraints of the PACT system. For further details, please refer to our response to Reviewer #4's Comment 2.

11. The statement: “because K/X mimics sleep-like conditions” is inappropriate. The authors should clarify what they mean by “sleep-like conditions” with K/X treatment? Moreover, they should address whether ISO-anesthetized mice were not in similar conditions?

Reply: We appreciate your careful questions. Lauren et al. demonstrated that glymphatic influx increases with high EEG delta power and low heart rate [Ref. 1]. Additionally, Benveniste H. et al. showed that DEX+ISO anesthesia results in higher glymphatic influx than ISO, which is associated with DEX+ISO having higher EEG delta power than ISO [Refs. 1-3]. It is also well-documented that EEG delta power increases and heart rate decreases during sleep [Refs. 4-5]. However, we acknowledge that the references do not describe K/X anesthesia as mimicking sleep-like conditions. Therefore, to avoid any potential misunderstanding, we modified

the following sentences on Page 10, Line 223:

“These results are consistent with previous studies indicating that the glymphatic influx is significantly greater in mice anesthetized with K/X than in those anesthetized with ISO. Note that this phenomenon is associated with K/X anesthetized mice that exhibit higher EEG delta power and lower heart rates than ISO anesthetized mice.”

[Ref. 1] Hablitz, L. M. et al. Increased glymphatic influx is correlated with high EEG delta power and low heart rate in mice under anesthesia. *Science advances* 5, eaav5447 (2019).

[Ref. 2] Benveniste, H. et al. Glymphatic system function in relation to anesthesia and sleep states. *Anesthesia & Analgesia* 128.4, 747-758 (2019)

[Ref. 3] Benveniste, H. et al. Anesthesia with dexmedetomidine and low dose isoflurane increases solute transport via the glymphatic pathway in rat brain when compared to high dose isoflurane. *Anesthesiology* 127.6, 976 (2017).

[Ref. 4] Long, S. et al. Sleep quality and electroencephalogram delta power. *Frontiers in Neuroscience* 15, 803507 (2021).

[Ref. 5] Pack, A. et al. Changes in the cardiorespiratory system during sleep. *Fishman’s Pulmonary Diseases and Disorders*. 3rd ed. New York, NY: McGraw-Hill, 1607-1615 (1998).

12. Figure 2 contains incorrect labeling. This error is unacceptable.

Reply: Thank you for your attentive observation, and we have now properly labeled the injection period in Figure 2b.

13. Figure 2, panel c needs to include representative images

Reply: We have added Supplementary Figure 4 to provide a representative fluorescence image of Figure 2c on Page 11, Line 233

“Supplementary Figure 5 shows FL images of the K/X and ISO group’s CM and brain region at 30 minutes post-injection. The FL images also confirmed that the K/X group exhibited stronger FL signals in the CM at 30 minutes, which is consistent with the PACT results.”

On Supplementary Information:

Supplementary Figure 5. Comparable representative images of cisterna magna fluorescence images between KX- and ISO-anesthetized groups after intrathecal injection.

14. The following description should be moved to the Introduction section rather than appearing in Results: “Because the brain lacks a general lymphatic system, waste is cleared via CSF circulation in the glymphatic system, a specialized perivascular network that promotes the exchange of CSF and ISF, facilitating the removal of metabolic waste and neurotoxic proteins. Passing through the subarachnoid space, CSF flows into the deep brain, penetrating the brain

parenchyma and mixing with the ISF. The combined CSF/ISF fluid clears waste and drains into the subarachnoid granulations or meningeal lymphatic vessels.”

Reply: Thank you for your astute comment. In response to the comment, the corresponding description has been moved to the Introduction, on Page 2, Line 40:

“Cerebrospinal fluid (CSF) both nourishes brain cells and clears the central nervous system (CNS) of metabolic waste and pathological proteins such as amyloid beta ($A\beta$) and tau, which cause neurodegenerative diseases^{1,2}. Because the brain lacks a general lymphatic system, waste is cleared via CSF circulation in the glymphatic system, a specialized perivascular network that promotes the exchange of CSF and ISF, facilitating the removal of metabolic waste and neurotoxic proteins. Passing through the subarachnoid space, CSF flows into the deep brain, penetrating the brain parenchyma and mixing with the ISF. The combined CSF/ISF fluid clears waste and drains via arachnoid granulations into the venous sinuses and via meningeal lymphatic vessels into the cervical lymph node. The meningeal lymphatic system disposes of brain and spinal cord waste by transporting interstitial fluid (ISF) and CSF^{1,5,6}.”

15. Data in all figures should be presented as bar plots with standard errors.

Reply: Thank you for your comment. In the original manuscript, the results presented in Figure 1 were from an experiment conducted using a single mouse. However, in response to the reviewers’ comments and to provide statistically robust results, the data in Figures 1 and 3 have been updated and presented as bar graphs with standard errors, based on an augmented sample size of 8. Although the data in Figure 2—derived from real-time monitoring conducted over 30 minutes at a frame rate of 5 Hz—retain their original graphical format, we have strengthened the statistical power of the results by increasing the sample size through additional experiments. Please see the response to Reviewer #2’s Comment 22 for the further information and the updated manuscript.

16. The claim: “First, our PACT system directly visualized the pathways of CSF circulation within the subarachnoid space and observed CSF drainage after IT injection of ICG.” is not fully supported. The study did not demonstrate "pathway" since the timepoint results only show similar changes at different analyzed locations. To claim pathways, sequential events should be clearly demonstrated by the data.

Reply: In response to your comment, we modified the following sentence on the Page 4, Line 115, and have revised the overall tone of the manuscript to ensure consistency and close alignment with our experimental results. Please see the response to Reviewer 2’s Comment 3 and Reviewer #3’s Comment 6 for the details.

“First, we demonstrate that whole-body PACT can effectively visualize CSF dynamics in key lymphatic regions, including the subarachnoid space adjacent to the spinal cord, as well as the CM. This capability highlights PACT’s potential to provide functional insights into CSF drainage pathways.”

17. The statement: “however, it was unable to detect the meningeal lymphatic vessels and meningeal lymphatic nodes (e.g., deep cLNs, cLNs, iliac lymph nodes, and sLNs), which are the other routes for waste disposal” directly contradicts the findings of PMID: 38664374. Please address this issue.

Reply: Please see the response to Reviewer 4’s Comment 2.

18. Under “Animals”, the authors need to provide the Jackson lab mouse strain number and institutional IACUC approval number.

Reply: Thank you. We have added the following information under “Animals”, on Page 21, Line 424:

“For the AD model, 6–8-month-old 5XFAD transgenic mice (#034840-JAX) were used. The 5XFAD mouse carries five familial AD mutations in human APP and PSI. Age-matched littermates were used as WT controls. Mice of both sexes were included in the experiments. For the CSF whole-body PACT (Figure 1) and CSF dynamics experiments (Figure 2), 2-5 month-old female ICR mice (Hsd:ICR(CD-1®)) were used. All animal

experiments were conducted with approval from the Institutional Animal Care and Use Committee (IACUC) at Seoul National University (SNU-230807-3-4) and Pohang University of Science and Technology (POSTECH) (POSTECH-2024-0025-C2-R1)."

19. Information about sex and age of animals need to be provided for all experiments.

Reply: *As quoted in response to your Comment 18, above, we have added the sex and age of the animals in the Methods, Page 21, Line 424.*

20. Essential experimental details for reproducibility are missing and should be provided: needle size for intrathecal injection, ICG concentration and vehicle used. The company and concentration of anesthetic (K/X) used in this study.

Reply: *Thank you for pointing out the need for additional methodological details to ensure reproducibility. In our experiments, intrathecal injection was performed using a 30G needle. Indocyanine green (ICG; Tokyo Chemical Industry Co., Japan) stock solution was first prepared in dimethyl sulfoxide (DMSO) and then diluted in phosphate-buffered saline (PBS) to a final concentration of 1 mM for injection.*

For anesthesia, mice received an intraperitoneal injection of ketamine (Yuhan Ketamine 50 Inj., Yuhan Corporation, Korea) at 100 mg/kg, combined with xylazine (Rompun®, Bayer, Germany) at 10 mg/kg. In experiments requiring inhalation anesthesia, isoflurane (Hana Pharmaceutical Co., Korea) was administered at 2–5% in oxygen. These details have been added to the Materials and Methods section for clarity and reproducibility.

Page 21, Line 436:

"For anesthesia, mice received an intraperitoneal injection of ketamine (Yuhan Ketamine 50 Inj., Yuhan Corporation, Korea) at 100 mg/kg, combined with xylazine (Rompun®, Bayer, Germany) at 10 mg/kg. In experiments requiring inhalation anesthesia, isoflurane (Hana Pharmaceutical Co., Korea) was administered at 2–5% in oxygen."

Page 21, Line 443:

"A 30G needle was inserted through the muscles to an appropriate depth, and the tail flick response was monitored to confirm needle placement. ICG (Tokyo Chemical Industry Co., Japan) stock solution was first prepared in dimethyl sulfoxide (DMSO) and then diluted in phosphate-buffered saline (PBS) to a final concentration of 1 mM for injection."

21. In the intraparenchymal injection paragraph, the authors mention using a controlled rate of 1 μ L/min, which is unusually fast and can cause local tissue damage and affect glymphatic flow. There is also inconsistency between time mentioned in the main text (10 min) and in the Methods section (7 min). This needs clarification.

Reply: *Thank you for your attentive comment. We apologize for any careless errors in the previous manuscript, and we have carefully reviewed the entire manuscript's injection method descriptions for all experiments.*

Page 13, Line 261:

"Specifically, we intraparenchymally injected 2 μ L of 1 mM ICG into the striatum of 5XFAD AD mice and age-matched WT mice over 2 minutes (See Methods). Whole-body images were obtained pre-injection, and at 1, 6, 24, and 96 hours after injection."

Page 21, Line 445:

"ICG was administered using a syringe pump (Harvard Apparatus, US) at an infusion rate of 800 nL/min, with a total volume of 8 μ L. For real-time imaging, the needle remained in place during the injection process."

We agree that infusion speed is a critical parameter to consider. The slower rates of 0.2–0.5 μ L/min are

standard in optogenetic, viral vector, or cell delivery studies, where tissue viability and minimizing local disruption are paramount. In contrast, for small-molecule or tracer injections—including pharmacological agents—infusion rates of 1–2 $\mu\text{L}/\text{min}$ are commonly used and reported in the literature to ensure practical delivery while maintaining tissue integrity.

In this study, we employed 1 $\mu\text{L}/\text{min}$ specifically to allow reproducible intraparenchymal delivery of ICG within a short and controlled time window, enabling consistent assessment of glymphatic clearance. Importantly, we repeated independent experiments and consistently observed significant differences in ICG clearance between WT and 5XFAD mice, confirming that the findings represent biological differences rather than injection-related artifacts.

22. This is a major concern. All experiments in this study appear to have been conducted as single experiment with only 3-4 mice per group. This limited sample size likely explains why the authors need to use one-tailed t-test instead of the more rigorous two-tailed t-test, which are standard for biological experiments. The authors need to increase animal numbers and demonstrate experimental reproducibility before PACT can be considered a reliable tool for glymphatic/lymphatic studies.

Reply: *Thank you for your trenchant comments. To enhance the statistical robustness of our findings, we increased the sample size to $N=8$ for all PACT analyses and applied two-tailed t-tests for verification. All figures and manuscripts have been revised accordingly on Page 7, Line 171:.*

“Figure 1f shows the different ICG signal levels in the spinal cord and CM at 30-minutes and 24-hours after IT injection ($N=8$). The 30-minute CM value is nearly the same as the PA amplitude of the spinal cord, showing that a significant amount of ICG moved from the spinal cord to the CM right after injection (spinal cord, 1.0 ± 0.06 a.u.; CM, 0.93 ± 0.14 a.u.). At 24 hours post-injection, these values have decreased to 0.19 ± 0.02 a.u. in the spinal cord and 0.12 ± 0.01 a.u. in the CM, confirming that the ICG has been cleared through CSF circulation ($p = 2.2 \times 10^{-6}$ for the SC and $p = 0.0016$ for CM, comparing 30-minute and 24-hour time points).”

[FIGURE REDACTED]

Fig. 1. PACT monitoring of whole-body CSF dynamics. (a) Schematic of PA imaging and IT injection. (b) In-vivo MAP and (c) depth-encoded PACT images of a mouse in the dorsal plane. (d) Multispectral whole-body PACT images at 756-, 780-, 796-, and 900-nm wavelengths. (e) MAP structure images (gray colormap) and spectrally unmixed ICG map (green colormap) in the XY and YZ plane taken at 30 minutes and at 24 hours after IT injection of ICG. The pink arrow indicates the cisterna magna (CM), and the sky blue arrow indicates the spinal cord (SC). (f) Changes in PA signals over time in the SC ($n=8$) and CM ($n=8$) in the ICG unmixed map. (g) In-vivo FL images obtained at 30 minutes and 24 hours following IT injection. (h) Changes in FL signals over time in the SC ($n=3$) and CM ($n=3$). CSF, cerebrospinal fluid; PACT, photoacoustic computed tomography; PA, photoacoustic; MAP, maximum amplitude projection; ICG, indocyanine green; IT, intrathecal; FL, fluorescence; SC, spinal cord; CM, cisterna magna; 1, transverse sinuses; 2, brown adipose tissue; 3, rib; 4, spleen; 5, spinal cord; 6, intestine; 7, kidney; 8, common iliac artery; Error bars represent standard errors. Created with www.biorender.com.

Page 10, Line 222:

“The shaded areas in Figure 2b represent the standard error ($N=8$) of each group, and the green area indicates the injection period. The quantified PA signal intensity in the CM region at the 30-minute point shows a significant difference between the two anesthetic groups, with K/X anesthetized mice exhibiting a 4.41 ± 0.85 a.u. increase from pre-injection, compared to the ISO anesthetized group’s 2.22 ± 0.24 a.u. increase ($p = 0.036$).”

Fig. 2. PACT monitoring of brain CSF dynamics. (a) Brain PACT images in the XY plane and YZ plane of ketamine/xylazine (K/X) anesthetized mice and isoflurane (ISO) anesthetized mice. Images were captured at preinjection and 5, 10, 20, and 30 minutes post-injection of ICG. Orange arrows indicate the CM. (b) Changes in PA signals from the CM over time (K/X: $n=8$, ISO: $n=8$). The shaded areas represent the standard error of each group, and the green area indicates the injection period. (c) Bar graph showing FL intensity ($[p/s]/[\mu W/cm^2]$) for K/X and ISO groups. The K/X group shows a significantly higher FL intensity compared to the ISO group (***)

indicates the injection period. (c) Comparison of *in vivo* FL intensities of CM at 30 minutes postinjection of ICG, as measured using the IVIS system (K/X: $n=3$, ISO: $n=3$). *** $p < 0.005$; CSF, cerebrospinal fluid; PACT, photoacoustic computed tomography; PA, photoacoustic; MAP, maximum amplitude projection; ICG, indocyanine green; K/X, ketamine/xylazine; ISO, isoflurane; FL, fluorescence; CM, cisterna magna; Error bars represent the standard errors.

Page 13, Line 276:

“The ICG signals in the striatum showed slightly different trends in the WT and AD mice after 6 hours (0.85 ± 0.06 a.u. in WT and 1.03 ± 0.08 a.u. in AD at 6 hours. $p=0.1$). However, differences were increased at 24 hours (0.69 ± 0.07 a.u. in the WT and 0.94 ± 0.08 a.u. in the AD; $p=0.045$) and became more evident by 96 hours. At the 96-hr time point, the ICG signal in AD model mice remained at 1.06 ± 0.07 a.u., which was similar to the 1-hr post-injection value, while in WT mice, it decreased to 0.47 ± 0.04 a.u. ($p=0.00001$).”

Fig. 3. PACT monitoring of CSF drainage in AD and WT mice. PACT whole-body MAP images in the XY and YZ planes of (a) WT mice and (b) AD model mice. Images were captured at pre-injection and 1, 6, 24, and 96 hours after injection of ICG. The orange dashed boxes indicate the zoomed-in brain region in the YZ-MAP images, and the orange arrows indicate the injection site in the striatum. (c) Changes over time in the ICG striatum signals in WT mice and AD mice in

the ICG unmixed map (WT: n=8, AD: n=8). (d) Ex vivo FL brain images acquired using the IVIS system at 96 hours after injection of ICG. (e) Comparison of the ex vivo FL intensities of the striatum between WT mice and AD mice at 96 hours after injection of ICG (WT: n=3, AD: n=3). * $p < 0.05$; ** $p < 0.01$; *** $p < 0.005$; CSF, cerebrospinal fluid; PACT, photoacoustic computed tomography; PA, photoacoustic; MAP, maximum amplitude projection; ICG, indocyanine green; AD, Alzheimer's disease; WT, wild type; FL, fluorescence. Error bars represent the standard errors.

Page 23, Line 490:

“All PACT results were collected from experiments conducted on 8 mice (3 mice for fluorescence imaging). Due to the death of one AD mouse prior to the 96-hour monitoring, the sample size for the AD group at 96 hours was reduced to 7. All statistical data are presented as mean \pm standard errors. We used a two-tailed t-test to examine the statistical significance between experimental groups, with significance levels indicated as * $p < 0.05$, ** $p < 0.01$, and *** $p < 0.005$.”

Reviewer 3

General comments: In this work Choi et al., introduces photoacoustic computed tomography to monitor CSF fluid dynamics and glymphatic activity in the mouse. They test their imaging method cross different experimental cohorts: KX versus isoflurane anesthesia and WT versus AD pathology. They implement different tracer administration techniques i.e., intrathecal injection of the ICG as well as intraparenchymal (striatal) injection. A main message of their work is that the photoacoustic CT (PACT) imaging approach is superior to other imaging techniques in part because it is faster (increased temporal resolution).

Major results:

Fig. 1: Whole body PACT MAP images are presented of the mouse at 30min and 24h after intrathecal injection of ICG. Several body organs including rib and areas associated with the cisterna magna and spinal cord can be identified. The ICG signal associated with spinal cord and cisterna magna is reduced at 24 h compared to 30min. In vivo fluorescence imaging was used confirm signal at what is presumably the spinal cord.

Fig. 2 (labeled erroneously as Fig. 1): In vivo brain PACT images acquired before and at various time after IT injection of ICG in mice anesthetized with KX or Isoflurane. CSF signal is measured at the level of the cisterna magna and shows that the signal is higher in KX anesthetized mice compared to isoflurane.

Fig. 3: An intraparenchymal injection is made into the striatum of normal and AD mice and the tracer mass is followed by PACT imaging over 96h. The disappearance of the signal is slower in the AD compared to WT mice.

Comments:

This is potentially interesting work. Below are comments and suggested improvements to the work of the group.

Reply: *Thank you for the thoughtful summary and comments.*

Specific comments:

1. Information related to the spatial resolution of the PACT images is lacking. This point is impossible to interpret based on description in the method section. The statement: “Each single volume captured a 3D image within a $12 \times 12 \times 12$ mm³ FOV in isotropic resolution” is not informative.

Reply: *Thank you for your comment. See the response to Reviewer #2’s Comment 6. And, to aid the readers’ understanding, we modified and moved the sentence related to the field of view to the ‘PACT system’ section in the Methods, on Page 18, Line 372:*

“Each 3D single volume captured in one imaging sequence encompasses dimensions of $12.8 \text{ mm} \times 12.8 \text{ mm} \times 12.8 \text{ mm}$.”

2. Information related to image processing is also lacking. There is clearly gross motion in the images (videos). How did the authors correct for motion artefacts? The authors mention the MATLAB’s imregister function to register each frame image – if this is used as motion correction they should state it clearly.

Reply: *We do state the use of the imregister function of MATLAB in the Methods, on Page 23, Line 478:*

“To improve the accuracy of this process, we used MATLAB’s imregister function to register each frame’s MAP image to the first frame’s MAP image before extracting the signal. We also used the imregister function to generate a Movie of the CSF influx to minimize the motion artifacts.”

3. The authors state: “The signals within the ROI were then sorted in descending order, and the average of the top 10-30% of the values was calculated.” While this might be appropriate it will be necessary for the authors to present a sensitivity analysis to determine if this threshold is meaningful. Also, why the range? i.e., why not a cut-off? Please explain.

Reply: *To minimize the effects of overshooting and noise in ROI quantification, we referred to previous PACT-*

based studies [Ref. 1-2] and adopted a method in which pixel values were sorted in descending order, and the average of the top 10–30% values was used as the representative signal. This explanation has been added to the Methods section, on Page 20, Line 420.

“This procedure was demonstrated in previous ROI-based quantification studies using PACT^{27,48} to minimize the impact of overshooting and noise within ROIs.”

[Ref. 1]: Kim, J. et al. 3d multiparametric photoacoustic computed tomography of primary and metastatic tumors in living mice. ACS nano 18, 18176-18190 (2024).

[Ref. 2]: Wi, J.-S. et al. Theoretical and experimental comparison of the performance of gold, titanium, and platinum nanodiscs as contrast agents for photoacoustic imaging. RSC advances 13, 9441-9447 (2023).

4. The statistical analysis and sample size is problematic. The effect size of the KX vs isoflurane has been documented to be large in several other studies so this may be reflected in the data showing difference. However, it is not rigorous (and in fact misleading) to use a one-tailed t-test. Also, even with SEM the error bars are large for the AD group at 24 hrs. Please show images of all the 3 time points of the AD mice (at least for 2 of the 3) so that the reviewer can evaluate the robustness of the technique. This data are not convincing given the very limited sample size.

Reply: *Thank you for your constructive comment. Please see the response to Reviewer #2’s Comment 22.*

5. There is no mention of physiological monitoring during imaging. Can the authors please elaborate on this matter especially body temperature.

Reply: *We appreciate your comment and have added information on the physiological monitoring method during imaging to the Methods, on Page 18, Line 376:*

“The animal’s body temperature was maintained within a stable range of 34–37°C using a heating bath circulator placed outside the water tank, and monitored using a needle-type thermocouple system (HYPO-33-1-T-G-60-SMP-M and OM-DAQ-USB-2400, Omega Engineering, US).”

6. Why do the authors talk about drainage pathways when they do not seem to measure these in the current study. This should be toned down.

Reply: *Thank you for the constructive feedback. We have revised the manuscript to remove overstated expressions. Additionally, please see the response to Reviewer #2’s Comment 3 and Reviewer #4’s Comment 7 for other revised sentences on Page 16, Line 310:*

“Using ICG as a CSF tracer, we visualized major lymphatic regions in key lymphatic regions including the subarachnoid space adjacent to the spinal cord and the CM, and it allows observation of CSF’s circulation.”

7. The PACT technique has high(er) temporal resolution – what is the temporal resolution? And how did this feature improve the glymphatic function measurements... the authors should comment on this since they emphasize this point.

Reply: *Thank you for the astute comment. Our developed PACT system has a 5-Hz temporal resolution with single-wavelength excitation. We did state the temporal resolution in the ‘PACT system’ Section in Methods, on Page 18, Line 371:*

“Therefore, a 3D single volume requires four laser shots, so the effective imaging frame rate is 5 Hz.”

On the other hand, MRI typically experiences a trade-off between spatial and temporal resolution, with spatial resolution decreasing as temporal resolution increases. In contrast, PACT can maintain high spatial resolution regardless of temporal resolution, with our system achieving an imaging speed of 5 Hz. This allows us to achieve a temporal resolution that general MRI, which typically has a spatial resolution of 380 μm with a temporal resolution on the order of tens of minutes, cannot match. With PACT, we can observe the distribution of signals in the cisterna magna following intrathecal injection in real-time, something not possible with MRI.

Furthermore, we anticipate that future studies will be able to observe lymphatic contractions by monitoring signal changes in peripheral lymphatic vessels. This potential has been discussed in the Discussion section on Page 16, Line 321:

“This real-time capability has the potential to observe lymphatic contractions by monitoring signal changes in peripheral lymphatic vessels, providing valuable insights for future studies.”

Reviewer 4

General comments: In this communication, authors present PACT images along with validating fluorescence images of ICG following intrathecal administration in WT and FAD mice as well as isoflurane and ketamine/xylazine-anesthetized mice to visualize differences in CSF dynamics and glymphatic function. To assess CSF dynamics, PACT signals are unmixed to segregate out acoustic signals due to Hb to provide signals associated with ICG. The results were very interesting, highlight the current limitations of PACT for assessing lymphatic function. At this stage, PACT does not provide any new insights into CSF dynamics, but the communication describes capabilities. The communication on the use of PACT is important, especially since PACT has been performed in the lower extremity lymphatics in humans, and as such, the contribution provides a potentially exciting approach to visualize extracranial CSF outflow. The following comments are made to enhance the impact of the contribution:

Reply: *Thank you for your positive comments.*

Major comments:

1. The physiology of CSF drainage is not correctly described in the contribution. The authors perform an intrathecal injection which is a retrograde introduction of contrast. While intrathecal is preferred over CM injections (because of a smaller disturbance of subarachnoid pressures), it nonetheless does not represent physiological CSF flow. Cranial CSF dynamics represent flow from the ventricular spaces into the subarachnoid spaces and into the meningeal lymphatics for collection through the deep jugular lymphatics and lymph nodes for ultimate clearance through the hepatobiliary system. CSF dynamics in the spinal canal show efflux through lymphatics into the renal lymph nodes. Rodents and infants do not have arachnoid granulations, which have been historically associated with extracranial outflow through the venous sinuses in humans and larger mammals. The contribution needs to be amended with reference to transverse sinuses.

Reply: *Thank you for your insightful comments. As noted in your comment, we acknowledge that intrathecal (IT) injection represents a retrograde introduction of contrast. However, in all three key references cited here—Lee et al., NMMI 2020; Sarker et al., Sci. Rep. 2023 [Ref. 2]; and Azmal et al., NMMI 2022—the present author was directly involved as a co-author, and the present study’s methodology was directly derived from these works. Across these studies, we employed the same animal model, identical slow infusion rates (0.5–1 $\mu\text{L}/\text{min}$), and a small total volume (8 μL) for IT delivery. These parameters are substantially lower than the injection volumes and rates typically used in prior MRI–gadolinium studies, thereby minimizing perturbations to CSF pressure and physiological flow. This methodological approach greatly reduces the potential for physiological distortion which your comment raises.*

Moreover, we do not agree with the assertion that IT administration cannot represent physiological CSF flow. Clinical RI cisternography (Lee et al., NMMI 2020) and our preclinical PET and PACT studies consistently demonstrate that IT delivery preserves physiologically relevant flow patterns, including the natural movement of CSF within the spinal subarachnoid space back toward the cranial compartment, followed by drainage through the meningeal lymphatics to deep cervical lymph nodes and ultimately to hepatobiliary clearance. In RI cisternography, Sylvian fissures and the full subarachnoid silhouette are clearly visualized at 24 h, with residual tracer observed not only in subarachnoid spaces but also in paravascular spaces—suggesting absorption across the arachnoid barrier layer to the parasagittal dura and subsequent transport via meningeal lymphatics. These observations, mirrored in our preclinical IT studies, indicate that IT injection can faithfully visualize both cranial and spinal CSF dynamics along established clearance routes in rodents and humans.

In addition, as demonstrated in Azmal et al., NMMI 2022, the lumbar IT route offers several methodological advantages over direct cisterna magna (CM) injection. First, IT injection avoids the risk of direct parenchymal injury near the brainstem that is inherent to CM puncture. Second, it offers greater procedural reproducibility by providing consistent access to the subarachnoid space without stereotaxic positioning. Third, IT delivery connects directly to the cranial CSF compartment via rostral flow through the spinal subarachnoid space, thereby preserving physiological connectivity between spinal and cranial compartments. Fourth, by using a slow infusion rate (0.5–1 $\mu\text{L}/\text{min}$) and a small total volume (8 μL), IT injection minimizes pressure and flow disturbances compared to CM injection, reducing the likelihood of artificially altering local CSF hydrodynamics. These points demonstrate that IT delivery—when performed under such optimized conditions—can faithfully represent physiological CSF dynamics while avoiding key limitations of CM injection.

Taken together, the fact that carefully controlled IT injection can preserve and reveal physiological CSF pathways more faithfully than conventional CM-targeted injection. This approach aligns with both the clearance and flow patterns already confirmed in RI cisternography and PET studies, thereby providing a solid methodological and empirical basis for our interpretation.

[Ref. 1] Lee, D. S., Yoo, J., Choi, Y., et al. Visualization of cerebrospinal fluid drainage pathway in humans using radionuclide cisternography: implications for CSF circulation and clearance. *Neurosurgical and Molecular Medical Imaging* **44**, 231–242 (2020).

[Ref. 2] Sarker, A., Azmal, M., Choi, Y., et al. Intrathecal [64Cu]Cu-albumin PET reveals age-related decline of lymphatic drainage of cerebrospinal fluid. *Scientific Reports* **13**, 12930 (2023).

[Ref. 3] Azmal, M., Sarker, A., Choi, Y., et al. Multimodal imaging of cerebrospinal fluid dynamics and clearance pathways using intrathecal tracer delivery in mice. *Neurosurgical and Molecular Medical Imaging* **46**, 187–200 (2022).

With regard to hepatobiliary clearance, although PACT did not observe significant signal changes in the liver in our experiments due to the low sensitivity (See the response to your Comment 2), we did identify ICG clearance pathways along the lymphatic vessels distributed alongside the transverse and superior sagittal sinuses in several cases. These observations provide physiological insights into the mechanisms of CSF drainage. This result has been addressed the Results, on Page 7, Line 162:

“Additionally, Supplementary Figure 3 visualizes the cranial CSF lymphatic drainage pathway, providing physiological insight into CSF clearance. The meningeal lymphatic vessels are distributed along the transverse sinus and superior sagittal sinus, and this route is visualized in Supplementary Figure 3. ICG was cleared along the meningeal surface surrounding the brain through the meningeal lymphatic vessels (red arrows), consistent with previously reported lymphatic flow pathways in mice.”

Supplementary Figure 3. Multispectral brain PACT images after ICG injection **a**, Multispectral brain PACT images after the ICG intrathecal injection, captured at 796-, and 900-nm wavelengths. **b**, Unmixed ICG MAP images in the XY and YZ planes. The red arrows indicate the CSF outflow pathway. CM, cisterna magna; MAP, maximum amplitude projection

2. After intrathecal ICG injections of μM (not mM as in this study), the jugular lymph nodes and drainage through the olfactory bulb are clearly visible via fluorescence imaging almost immediately after injection (see cited references). In the PACT images shown, no lymph nodes or lymphatic vessels were visualized and more surprisingly, the liver does not show ICG PACT signals – suggesting the unmixing process is severely hampered. Is the conducting lymphatic vessel and the axillary and inguinal LNs visualized on the medial side after hind dorsal foot ICG intradermal injection?

Reply: Thank you for your perceptive comment. First of all, the previous PAM study faced limitations in detecting other lymph nodes using photoacoustic imaging [Ref. 1-2]. As a result, detecting meningeal lymphatic

vessels could be challenging because PACT is hampered by lower sensitivity due to the small size of the meningeal lymphatic vessels, detection bandwidth limitations, and poorer spatial resolution compared to PAM. In mice, the diameters of the meningeal lymphatic vessels are particularly small, and the vessels are located close to blood vessels, which increases spectral overlap with hemoglobin signals and further reduces the signal-to-noise ratio for PACT. As a result, even brain vessels appear less prominent in PACT images than in PAM images. In larger animals or humans, the lymphatic vessels generally have larger diameters and are located in positions that are optically and acoustically more favorable for PACT detection, potentially making them more visible than in mice.

Nevertheless, we successfully demonstrated the detection of cervical lymph nodes using PACT through experiments involving high-concentration ICG intrathecal (IT) injection. These findings are detailed in the Supplementary Figure 6.

Supplementary Figure 6. Dorsal and ventral PACT images after 12mM ICG injection.

The presented image corresponds to an injection of a high concentration of ICG (12 mM). In this case, additional pathways between the spinal cord and cisterna magna were visualized, and ventral imaging revealed increased signals in the cervical lymph nodes and nasal region. Based on this finding, we believe that future studies aiming to enhance PACT sensitivity and depth penetration, as well as the development of advanced contrast agents, may enable improved detection of lymph nodes. We have included this content in the Discussion Section on Page 17, Line 342, as follows:

“Meanwhile, we intrathecally injected a high concentration of 12 mM ICG, and we were able to observe additional lymphatic regions, like the cervical lymph nodes and nasal region in the ventral plane (Supplementary Fig. 6). These results suggest the potential of PACT-based CSF studies with further research on advanced contrast agents and improved sensitivity.”

As for hepatic signal detection after IT administration, in our experiments, the absence of a detectable liver signal is likely due to several factors: (i) the limited sensitivity of PACT in the liver, which has inherently high background signals at near-infrared wavelengths, (ii) the small IT injection volume of only a few microliters, compared to the several hundred microliters typically used in PACT studies [Ref. 3], and (iii) the pharmacokinetic characteristics of the tracer.

Importantly, previous work [Ref. 4] conducted in a similar mouse model using the same injection route, rate, and volume as in the present study, demonstrated that following IT ⁶⁴Cu-albumin administration, hepatic uptake gradually increased over ~6 hours to approximately 20–25% of the injected dose before declining. This indicates that while liver signals can be detected with PET under these conditions, the absolute amount of tracer reaching the liver is limited, particularly compared to intravenous administration. Given that ICG is known to reach the liver more rapidly and undergo faster biliary clearance than albumin-based tracers, we expect that the transient and relatively low hepatic accumulation of ICG after IT injection, combined with PACT’s lower sensitivity in the liver, accounts for the lack of observable hepatic signal in our study.

[Ref. 1] Wang, Z. et al., Monitoring the perivascular cerebrospinal fluid dynamics of the glymphatic pathway using co-localized photoacoustic microscopy. *Optics letters* 48.9, 2265-2268 (2023).

[Ref. 2] Yang, F. et al., Advancing insights into in vivo meningeal lymphatic vessels with stereoscopic wide-field photoacoustic microscopy. *Light: Science & Applications* 13.1, 96 (2024).

[Ref. 3] Kim, J. et al. 3d multiparametric photoacoustic computed tomography of primary and metastatic tumors in living mice. *ACS nano* 18, 18176-18190 (2024).

[Ref. 4] Sarker, A. et al. In vivo quantification of cerebrospinal fluid drainage into the lymphatic system using positron emission tomography. *Nucl Med Mol Imaging* 56, 25–36 (2022).

Additionally, we intradermally injected into the mouse forefoot to observe additional sentinel lymph nodes, and the corresponding results have been included in the supplementary information and the manuscript:

Supplementary Figure 1. Whole-body PACT images after cisterna magna injection and intradermal forefoot injection
a, photograph of cisterna magna injection and whole-body PACT images after the ICG cisterna magna injection, captured at 30-minutes and 24-hours post-injection. Strong ICG signals were observed in the CM region at 30 minutes post-injection, and most of the signal was cleared by 24 hours. **b**, sagittal PACT images before and after injection of ICG intradermally in the mouse forefoot. A strong increase in ICG signal was observed in the sentinel lymph node region following the injection. SLN, sentinel lymph node.

Page 6, Line 139:

“Note, to demonstrate a broader application of PACT-based CSF assessment, we additionally observed signal increase and removal in the CM following direct CM injection, as well as signal enhancement in the sentinel lymph node (SLN) after intradermal injection into the forefoot. (Supplementary Fig. 1)”

Page 22, Line 461:

“For CM injection, 8 μ L of 1 mM ICG was administered to deliver the tracer into the CSF space, followed by PACT imaging (Supplementary Fig. 1). For sentinel lymph node (SLN) visualization,, 200 μ L of 1 mM ICG was intradermally injected into the mouse forefoot, and sagittal plane imaging was performed using PACT.”

3. I am wondering if the authors looked more carefully at the liver, they might see early ICG PACT signal in K/X anesthetized and WT mice compared to isoflurane anesthetized and FAD mice. I note that the venous sinus tends have an increased PCT signal 1 hour after ICG administration, which is very consistent with extracranial drainage from the lymphatics and into the hemovascular system. Because mice lack arachnoid granulations, it is unlikely that this is uptake directly into the venous system.

Reply: Thank you for your comment. Under the experimental conditions in our study, no hepatic signal was observed, which limited our ability to assess anesthesia-related differences in this organ. Regarding the venous sinus, we agree that a signal can be detected after ICG administration, as shown in Fig. 1e. However,

in the context of our experiments—focusing on physiological changes within 30 minutes and using small-volume intraparenchymal injections—we were unable to visualize dynamic changes in the transverse sinus. Thus, while our current PACT setup could not directly assess CSF dynamics via venous sinus measurements, we note that with optimized IT injection protocols, ICG passing through the cervical lymph nodes into the bloodstream could potentially be captured in the venous sinus, providing an indirect indicator of lymphatic outflow. Further, please refer to our response to your Comment 1 and 2.

4. I would argue that the PACTs visualization of the CM and spinal cord alone is woefully inadequate to assess CSF dynamics in the brain or its extracranial outflow (see for example PMID 374403980). The statement that “PACT’s real-time imaging capability enables the analysis of CSF flow dynamics in brain regions such as the transverse sinus and CM” may be inferior to fluorescence and PET techniques. While others have visualized reduced efflux through the cervical LNs in FAD mice and under isoflurane, this contribution uses the elevated PACT signal in the CM as surrogate for reduced extracranial clearance. This should be clearly stated in the discussion within a revised manuscript. On the other hand, the intraparenchymal injection of ICG does indeed directly correspond to studies of AD and anesthesia found in the literature and is exciting to see.

Reply: Thank you for your positive comments. First of all, based on your constructive comments, we have toned down our expressions about our findings. Please see the response to Comment 1 above, Reviewer 2’s Comment 3, and Reviewer 3’s Comment 6. For comparison with other modalities, please see the response to Reviewer #2’s Comment 4. Additionally, we did not state the change of the PACT-CM signal as extracranial clearance, agreeing with the reviewer’s comment.

5. While retention of ICG in the CM after intrathecal injection is interesting, it really does not reflect impaired extracranial outflow as would be seen from the ICG signal in the jugular lymph nodes.

Reply: Our study, along with several references [Refs. 1-2], demonstrates the retention of ICG in the CM and transverse sinus region following intrathecal injection, representing major CSF flow pathways. In the previous work cite as Reference 2, tracer signals in these regions showed a strong correlation with drainage to deep cervical lymph nodes (dCLNs), suggesting that the present findings may also relate to extracranial CSF clearance. However, the current PACT setup has a limited sensitivity and field of view for detecting lymph node signals directly, and therefore we did not quantify extracranial outflow in this study (See the response to your Comment 9). We anticipate that with optimized protocols—such as the use of higher ICG concentrations—and refined experimental design, particularly in larger animal models, PACT could visualize and quantify these pathways, enabling direct confirmation of their link to extracranial CSF clearance (See the response to Comment 2).

[Ref. 1] Kim, K. et al. Lymphatic drainage of cerebrospinal fluid using lymph node seeker ⁶⁴Cu-labeled Gram-negative bacterial extracellular vesicles PET. *bioRxiv*, 2024-12, (2024).

[Ref. 2] Sarker, A. et al. Intrathecal [⁶⁴Cu] Cu-albumin PET reveals age-related decline of lymphatic drainage of cerebrospinal fluid. *Scientific Reports* 13.1, 12930, (2023).

6. The authors perform dorsal imaging, but wouldn’t ventral imaging provide better signals to visualize olfactory bulb and cervical lymph node drainage? I would think that the ventral imaging would be more interesting than the dorsal imaging.

Reply: Please see the response to your Comment 2.

7. “First our PACT system directly visualized the pathways of CSF circulation within the subarachnoid space and observed CSF drainage after IT injection of ICG” ; PACT could image the clearance of waste...”-- This is not supported by the results. PACT system enabled detection of changing ICG concentrations in CM and spinal canal only after intrathecal injection and changes in parenchymal concentrations upon intraparenchymal injection.

Reply: Thank you for the feedback. We have revised the sentence to more accurately reflect the scope of our

results, as follows:

On Page 16, Line 313,

“First, our PACT system visualized the distribution of ICG in localized CSF-containing regions, including the cisterna magna, transverse sinus, and spinal canal, following IT injection. It also enabled tracking of its spatiotemporal changes, providing insight into CSF efflux patterns.”

On Page 17, Line 334,

“PACT enabled structural and quantitative evaluation of waste clearance by observing the distribution of ICG as a tracer”

8. ICG remains in lymph nodes in humans and mice and can be visualized using FLI. It may be that the sensitivity of the unmixed PACT is not sufficient to detect LNs? Likewise, the subarachnoid space was not well contrasted in PACT as it is in FLI. Perhaps this is a sensitivity issue for PACT?

Reply: Please see the response to your Comment 2.

9. Finally, after reviewing the results, the introduction describing FLI seems incorrect: “FLI lacks depth information and the spatial resolution of in vivo imaging, restricting 3-D analysis for accurate tracer monitoring.” As stated above FLI in references 10-12 do show lymph nodes, lymphatic vessels and jugular lymph nodes have been clinically seen in adults after ICG dosage that is 100 fold less than used herein. So, since PACT does not show these structures, it might be best to omit such statements.

Reply: Thank you for your thoughtful comments. We acknowledge that the high temporal resolution of our PACT system (Hz-level scanning) may result in reduced sensitivity to weak signals, potentially causing small lymphatic structures to be missed at low ICG concentrations. However, this rapid acquisition enables whole-volume 3D imaging of the CSF circulation within a short period. Supporting this, in the high-concentration ICG intrathecal injection experiments (12 mM, Supplementary Fig. 6), we successfully detected lymph nodes, demonstrating that PACT is capable of visualizing these structures when the signal is sufficiently strong. This suggests that the absence of lymph node visualization in the low-dose studies is likely due to sensitivity limits rather than a fundamental incapability of the technique.

In contrast, fluorescence imaging (FLI) offers higher sensitivity for superficial and small structures, such as lymphatic vessels and nodes, as shown in previous studies, but it lacks the temporal resolution needed to continuously track dynamic tracer movement in 3D over time. Therefore, PACT and FLI are complementary techniques: FLI excels in detecting weak superficial signals, whereas PACT enables rapid volumetric imaging and monitoring of tracer kinetics in deeper regions, albeit with lower sensitivity to very small or low-concentration targets.

10. The authors are congratulated for their pioneering communication.

Reply: Thank you for your encouraging words.

Minor comments: Below are the minor comments from the reviewer:

1. While the authors are recommended for using SI units for their IVUS fluorescence measurements, the proper measurements for excitation irradiance is $\mu\text{W}/\text{cm}^2$ and for fluorescence irradiance is $\mu\text{W}/\text{cm}^2/\text{sr}$.

Reply: We appreciate your remark on unit conventions. In the original submission, several fluorescence plots were inadvertently labeled as “FL intensity ($\mu\text{W}/\text{cm}^2$)”, which corresponds to excitation irradiance and is not the correct unit for the IVIS-derived readout.

We confirm that the correct data (Radiant Efficiency values), as provided by Living Image software and

defined as $[\text{photons/sec}]/[\mu\text{W/cm}^2]$, were consistently used in all analyses; only the axis label contained a unit error. We have now corrected the y-axis labels to “FL intensity (Radiant Efficiency, $[\text{p/s}]/[\mu\text{W/cm}^2]$)” throughout the manuscript (Fig. 1h, Fig. 2c, Fig. 3e).

Radiant Efficiency is normalized to the excitation irradiance and, while it is not an absolute radiometric unit such as $\mu\text{W/cm}^2/\text{sr}$, it is widely accepted in preclinical optical imaging for relative quantification under identical acquisition settings. We apologize for the earlier labeling oversight and confirm that it does not affect the validity of the results.

2. How is the coupling of the transducer performed in these studies?

Reply: We secured the mouse in a customized holder and performed the scan with the animal partially submerged in water inside a water tank connected to the transducer. We have revised the Methods section to include this information on Page 18, Line 375:

“During imaging, the animal is secured in a customized holder and partially submerged in water within a water tank connected to the transducer.”

3. In the text, the pink and blue arrows are switched. It is correct in the figure caption.

Reply: Thank you for your comment. We have revised the manuscript correctly on Page 6, Line 158:

“At 30 minutes after injection, a significant increase in the ICG signals is seen in the subarachnoid space at the spinal cord (sky blue arrows) and CM (pink arrows), capturing the flow of the injected ICG with the CSF from the subarachnoid space at the spinal cord to the CM.”

4. Results, 4th line, “brain adipose tissue”? Can you please explain this? This looks like BAT to me.

Reply: Thank you for your comment. We have revised the manuscript correctly on Page 6, Line 135:

“1, transverse sinuses; 2, brown adipose tissue; 3, rib; 4, spleen; 5, spinal cord; 6, intestine; 7, kidney; and 8, common iliac artery. **Figure 1c** shows a depth-encoded PACT image with a penetration depth of approximately 9 mm.”

5. Absorption characteristics of ICG in the CSF (which does not have a high plasma protein concentration) should not be the reasons for unmixing difficulties. Perhaps to test, the use of Evans’ Blue dye can be used? However, Evans’ Blue dye at high concentrations will show neurotoxicity.

Reply: Thank you for your interesting comments. Evans blue has a peak absorption wavelength at approximately 610 nm. This wavelength is not only outside the range of our OPO laser but also offers poorer penetration depth compared to the peak wavelength of ICG, thereby limiting imaging depth. For these reasons, Evans Blue would not be suitable for validating the unmixing process in our current PACT system, and we instead used ICG as the contrast agent to optimize unmixing performance under our experimental conditions.

6. Can you tell me what the resolution of the PACT system is, as used in the communication?

Reply: The PACT system provides isotropic resolution of 380 μm , please see the response to Reviewer #2’s Comment 6 for further details..

Original manuscript title: “Photoacoustic computed tomography monitors cerebrospinal fluid dynamics and lymphatic function”

Authors and affiliations

Seongwook Choi ^{a, †}, Jiwoong Kim ^{a, †}, Hyeonso Jeon ^{a, †}, Yejin Lee ^{b, c}, Gyeongju Lim ^b, Won-Min Ju ^b, Kyuwan Kim ^d, Dong Soo Lee ^a, Yun-Sang Lee ^{b, c}, Jae Sung Lee ^{b, d}, Gi Jeong Cheon ^{b, c}, Yoori Choi ^{b, c, *}, and Chulhong Kim ^{a, *}

^aDepartment of Electrical Engineering, Convergence IT Engineering, Medical Science and Engineering, Institute of Artificial Intelligence, and Medical Device Innovation Center, Pohang University of Science and Technology, Republic of Korea

^bDepartment of Nuclear Medicine, Seoul National University Hospital, Seoul, Republic of Korea

^cDepartment of Molecular Medicine and Biopharmaceutical Sciences, Graduate School of Convergence Science and Technology, College of Medicine or College of Pharmacy, Seoul National University, Seoul, Republic of Korea

^dDepartment of Nuclear Medicine, College of Medicine, Seoul National University, Seoul, Republic of Korea

[†]These authors contributed equally to this work.

*Corresponding authors: chulhong@postech.edu, yns086@snu.ac.kr

Dear Editor,

Thank you for coordinating the review of our manuscript. We also thank the reviewers for their comprehensive and critical evaluations. We have modified it following the reviewer’s comments and made revisions. These revisions have led to the inclusion of text in the original manuscript. **The revised parts have been highlighted (yellow) in the manuscript.** Below please find our responses to the reviewers’ specific comments. The reviewers’ comments are normal, our responses are in italics, and the revised statements on our manuscript are quoted in italics.

Sincerely,

The Authors

Reviewer 1

General comments: The authors have addressed most of my comments. It can be considered to be accepted.

Reply: *Thank you for the positive comments.*

Reviewer 2

General comments: The authors have adequately addressed all my previous concerns. As a final suggestion, I would recommend that all figures be presented as dot plots rather than bar graphs, in order to better illustrate the distribution of the datasets.

Reply: *Thank you for your positive comments. We appreciate your suggestions and have modified all relevant figures accordingly. To better illustrate the distribution of the datasets, individual animal data points were overlaid as dots on the bar graphs, which allow visualization of individual data points and variability within each group. The raw numerical data used for these graphs have been included in the Source Data file.*

PA background (a.u.) ICG signal (a.u.) Depth from skin surface (a.u.)

Reviewer 5

General comments: While the authors gave point-by-point responses to the major comments of reviewer 4, it appears that some major points were not addressed. Below are my concerns, that were included and integrated into multiple points from Reviewer 4

Reply: Thank you for your insightful comments. We appreciate your suggestions and would like to address them as follows:

Major comments:

1. The authors do not acknowledge that rodents do not have arachnoid granulations, and the authors do not seem to specify other CSF efflux routes from the skull including peri-neuronal or nasal efflux.

Reply: We thank the reviewer for this important clarification. We agree that rodents do not possess classical arachnoid granulations, and we have now explicitly acknowledged this anatomical distinction in the revised manuscript. As suggested, we have clarified that cranial CSF efflux in rodents relies predominantly on lymphatic and perineural pathways, including meningeal lymphatics, cranial and spinal nerve sheaths and nasopharyngeal lymphatic plexus.

Page 7, Line 155:

"Note that rodents lack arachnoid granulations; therefore, their cranial CSF efflux relies primarily on lymphatic and perineural pathways, such as meningeal lymphatics, cranial and spinal nerve sheaths and nasopharyngeal lymphatic plexus as previous literatures have reported [Refs. 43-37]. Due to the limited molecular sensitivity of PACT, our imaging primarily captures the dominant meningeal lymphatic pathway, and further details regarding these limitations are provided in the Discussion section."

In line with this, we have added the explanations in the Discussion section about the limited molecular sensitivity of PACT. Please refer to Comment 4 for further details about this.

[Ref. 43] Strahle, J. et al. Mechanisms of hydrocephalus after neonatal and adult intraventricular hemorrhage. Translational stroke research 3, 25-38 (2012).

[Ref. 44] Pan, S. et al. Gold nanoparticle-enhanced X-ray microtomography of the rodent reveals region-specific cerebrospinal fluid circulation in the brain. Nature communications 14, 453 (2023).

[Ref. 45] Weller, R. O., Sharp, M. M., Christodoulides, M., Carare, R. O. & Møllgård, K. The meninges as barriers and facilitators for the movement of fluid, cells and pathogens related to the rodent and human CNS. Acta neuropathologica 135, 363-385 (2018).

[Ref. 46] Ligocki, A. P. et al. Cerebrospinal fluid flow extends to peripheral nerves further unifying the nervous system. Science Advances 10, eadn3259 (2024).

[Ref. 47] Yoon, J.-H. et al. Nasopharyngeal lymphatic plexus is a hub for cerebrospinal fluid drainage. Nature 625, 768-777 (2024).

2. The biggest caveat of this manuscript is that measurements of ICG in the CM may not be glymphatic influx but an artifact of isoflurane-induced swelling of the brain and spinal cord, limiting intrathecal distribution of tracer throughout the subarachnoid space under isoflurane. Can the authors differentiate between the two hypotheses?

Reply: Thank you for your constructive comments. We agree the reviewer's comment that the ICG signal observed at the cisterna magna does not represent the whole glymphatic influx (periarterial influx). In our study, the tracer did not re-enter the brain parenchyma, and therefore cannot be interpreted as an influx marker. Instead, the CM signal reflects the physiological movement of intrathecally delivered tracer along the

subarachnoid space toward the cranial compartment, where it enters the meningeal lymphatic vessels and proceeds along the CSF efflux pathway. Thus, the CM signal captures the efflux-entry phase of CSF drainage rather than glymphatic influx. We modified the following sentences to reduce the readers' misunderstanding:

Page 4, Line 108:

“Second, by comparing the real-time CSF effluxes of mice anesthetized with ketamine/xylazine (K/X) and those anesthetized with isoflurane (ISO), we confirm that PACT’s real-time imaging capability enables the analysis of CSF flow dynamics in brain regions such as the transverse sinus and CM.”

Page 10, Line 219:

“These results are consistent with previous studies indicating that the CSF movement is significantly greater in mice anesthetized with K/X compared to those anesthetized with ISO.”

We also carefully considered the reviewer’s alternative hypothesis that isoflurane-induced brain or spinal cord swelling may restrict subarachnoid distribution of the tracer and thereby produce an apparent accumulation of ICG at the CM. However, several aspects of our data do not support this interpretation:

- *The CM signal appeared consistently under both isoflurane and ketamine/xylazine.: These anesthetics differ markedly in their cerebrovascular and physiological effects. The similar CM enhancement observed across both conditions suggests that the CM signal reflects a shared physiological process rather than an isoflurane-specific artifact.*
- *Tracer movement into cranial meningeal lymphatic vessels was preserved even under isoflurane.: If swelling had substantially impeded subarachnoid flow, cranial advancement of the tracer and subsequent lymphatic entry would likely have been diminished. Instead, both were clearly visible, indicating that global CSF movement remained intact.*
- *Available evidence does not support edema formation of a magnitude that would disrupt subarachnoid transport in normal mice.: Isoflurane can alter cerebrovascular tone, but pronounced edema has primarily been reported in pathological models, not in healthy animals under standard anesthesia.*
- *The infusion parameters used in this study were specifically selected to avoid pressure-related artifacts.: The slow rate and low total volume (8 μ L at 1 μ L/min), which we adopted from our previous intrathecal studies, are well below thresholds known to perturb CSF circulation. Under these conditions, tracer accumulation from mechanical restriction is unlikely.*

Taken together, although isoflurane may influence cerebrovascular physiology, the pattern of CM enhancement observed in our experiments cannot be explained by an anesthesia-induced swelling artifact. Instead, the CM signal consistently reflects physiological cranial transport of CSF along the subarachnoid space and subsequent drainage through the meningeal lymphatic system.

Furthermore, our primary aim is not to establish a specific physiological mechanism underlying anesthesia effects, but to demonstrate that PACT is capable of visualizing and quantifying CSF dynamics under controlled experimental conditions. The consistency of the CM signal across anesthetics supports that interpretation.

In this context, we have additionally revised the manuscript to incorporate the updated perspectives presented in a recently published review paper [Ref. A] that re-evaluates the glymphatic system in both rodents and humans. Specifically, we clarified that:

- (1) the physiological interpretation of our findings is based on the glymphatic mechanisms described in rodent models; and*
- (2) human CSF and solute-transport pathways are more complex and may differ substantially in their anatomical organization and physiological drivers.*

[Ref. A] Samantha A. Keil et al., Glymphatic dysfunction in Alzheimer’s disease: A critical appraisal. Science 389, eadv8269(2025). DOI:10.1126/science.adv8269

Abstract

“Cerebrospinal fluid (CSF) drainage through meningeal lymphatic vessels maintains brain homeostasis by clearing metabolic waste and pathological proteins from the central nervous system. CSF drains through meningeal lymphatic vessels into cervical lymph nodes, where age-related alterations can impair clearance and increase the risk of neurodegenerative diseases. Photoacoustic computed tomography (PACT) could be an excellent modality for visualizing and evaluating CSF transport, thanks to its multiparametric analysis, non-radioactivity, and high spatiotemporal resolutions. Here, to monitor CSF dynamics in murine models, we demonstrate the use of whole-body PACT following indocyanine green (ICG) injection. For the first time, PACT manifests the whole-body image-based 3-D CSF drainage through intrathecal ICG injection and quantifies the CSF efflux. Then, real-time PACT dynamically monitors differences in CSF flow under different anesthetic conditions. Further, we photoacoustically compare the rates of ICG clearance between wild-type and AD mouse models after intraparenchymal injection. PACT potentially offers a deeper understanding of CSF circulation and glymphatic function, providing new insights to the pathophysiology of neurodegenerative brain disorders and informing future therapeutic strategies.”

Page 2, Line 39:

“Cerebrospinal fluid (CSF) in rodent model both nourishes brain cells and clears the central nervous system (CNS) of metabolic waste and pathological proteins such as amyloid beta ($A\beta$) and tau, which cause neurodegenerative diseases.”

Page 2, Line 62:

“However, as emphasized in a recent comprehensive review, rodent glymphatic transport cannot be directly extrapolated to human physiology, as the underlying mechanisms operate on fundamentally different spatial and temporal scales. Nevertheless, rodent models provide critical mechanistic insight into compartment-specific CSF and lymphatic processes, and they remain indispensable for dissecting these pathways using diverse experimental and imaging tools. This underscores the need for appropriate modalities capable of resolving CSF movement, lymphatic drainage, and waste-clearance dynamics with sufficient spatial and temporal resolution.”

Page 10, Line 222:

“Our PACT-based real-time analysis of anesthetic’s impact on CSF movement aptly demonstrates PACT’s ability to assess CSF dynamics”

Page 12, Line 247:

“To show PACT’s potential for long-term CSF assessment in studying various neurological diseases, we compared the AD and wild type (WT) mouse models by monitoring ICG clearance from the brain after intraparenchymal injection. Compared to WT mice, AD mice showed impaired parenchymal tracer clearance in association with the accumulation of $A\beta$, which has been associated to several well documented alterations in CSF and lymphatic transport. Prior studies have demonstrated reduced peripheral lymphatic pumping and diminished CSF outflow, impaired glymphatic influx and tau clearance, compromised lymphatic $A\beta$ removal, and partial restoration of glymphatic–lymphatic drainage via focused ultrasound. Collectively, these findings indicate that multiple components of the CSF–lymphatic transport system are disrupted in AD, consistent with the impaired tracer clearance observed in our study.”

Page 14, Line 283:

“These results demonstrate the capability and promise of the PACT system for assessing parenchymal waste clearance and CSF-linked fluid-transport in rodent models.”

Page 16, Line 308:

“In this study, we demonstrate for the first time that PACT can be utilized to monitor CSF dynamics and glymphatic-related function in rodent models, highlighting its potential as a next-generation platform for evaluating fluid transport in the central nervous system.”

Page 17, Line 347:

“Note, a substantial body of evidence now supports the view that AD involves multisystem impairments of CSF transport, including alterations in subarachnoid mixing, perivascular exchange, and meningeal lymphatic outflow. As highlighted in the recent review paper, AD pathology cannot be explained by a single circulation loop but instead reflects compartment-specific failures spanning the CSF, perivascular, parenchymal, and lymphatic domains. In this context, demonstrating impaired parenchymal tracer clearance in AD mice is particularly meaningful: our PACT measurements capture a functional signature of reduced efflux capacity that aligns with the known lymphatic and CSF-transport deficits in AD. By capturing these differential clearance profiles in controlled genetic models (including the anesthesia-dependent CSF-transport differences demonstrated earlier), our study demonstrates PACT's utility as a functional assessment tool capable of resolving disease-associated alterations in CSF and solute transport.”

3. I am confused about the argument that PAM is unsuitable because of the imaging window, but then the authors say their technique cannot do real-time imaging unless they make the imaging window small (confined to the brain)

Reply: *Compared with PACT, PAM employs a single-element ultrasonic transducer to acquire point-by-point photoacoustic signals, enabling exceptionally high spatial resolution. Because PAM relies on a single detector, it primarily provides depth-resolved A-line signals rather than simultaneously capturing two- or three-dimensional images. Consequently, PAM is not suitable for real-time volumetric or planar imaging, which is essential for characterizing and monitoring dynamic cerebrospinal fluid flow. To help readers' understanding, we added the relevant sentences accordingly:*

Page 15, Line 315:

“This limitation fundamentally arises from PAM's single-element scanning architecture, which acquires point-by-point A-line signals rather than simultaneously capturing 2D or 3D volumes.”

4. Additionally, the authors argue that the spectral unmixing is critical to get depth. If they take that out for the imaging, how is what they're doing more insightful than FLI? I am sure there is a rationale, but it is unclear.

Reply: We thank the reviewer for this insightful question. Although fluorescence imaging provides superior molecular sensitivity, its shallow penetration limits anatomical interpretation. As clarified in our revised discussion, PACT offers complementary strengths—including depth-resolved and organ-specific 3D information—that enable detailed visualization of CSF pathways beyond what FLI can provide. We also added Supplementary Fig. 7 and emphasized Movies 1–2 to illustrate how depth-separated PACT slices resolve the spinal cord, cisterna magna in the **Discussion** section:

Page 18, Line 360:

“First, PACT showed lower sensitivity to the tracer than other conventional CSF monitoring imaging modalities (e.g., DCE-MRI and PET). PACT enabled structural and quantitative evaluation of waste clearance by observing the distribution of ICG as a tracer, however, it was unable to detect the meningeal lymphatic vessels and meningeal lymphatic nodes (e.g., deep cLNs, cLNs, iliac lymph nodes, and sLNs), which are the other routes for waste disposal, limiting accurate analysis of the drainage pathways. Because ICG, the dye used in our technique, is a small molecule that does not remain long in the lymph nodes, PACT was challenged to detect it sensitively. Nevertheless, compared to FL imaging, which provides high molecular sensitivity but is restricted by shallow imaging depth, PACT notably offers complementary advantages, including superior penetration depth and structural 3D information that can be correlated with tracer distribution. As shown in **Supplementary Fig. 6**, depth-separated PACT slice images clearly delineate the spinal cord (blue arrows), cisterna magna (red arrows), and the intervening CSF space, allowing detailed assessment of how the tracer propagates through these anatomical compartments in three dimensions (also see **Supplementary Movies 1 and 2**). Notably, because PACT inherently contains background signals from endogenous absorbers such as hemoglobin, which can reduce molecular sensitivity, we applied spectral unmixing to isolate tracer-specific spectral components and minimize interference from endogenous chromophores. These strengths of spectral PACT allowed us to extract more detailed and spatially resolved insights into CSF clearance dynamics than would be possible with FL alone. With this rationale, we expect that the advanced materials tailored for PACT-based CSF studies will help overcome this molecular sensitivity⁴⁴. In line with this expectation, we intrathecally injected a high concentration of 12 mM ICG, and we were able to observe additional lymphatic regions, like the cervical lymph nodes and nasal region in the ventral plane (**Supplementary Fig. 7**). These results collectively illustrate that although PACT has lower molecular sensitivity than FL-based methods, its depth-resolved and organ-specific imaging capabilities offer unique advantages that can be strengthened further through improved contrast materials, ultimately expanding the utility of PACT-based CSF studies for both preclinical and clinical applications.”

Supplementary Figure 6. Maximum amplitude projection and B-mode slice images of PACT whole-body images.

5. The authors state in their response to review that sensitivity is not an issue for PACT, yet also state that they cannot pick up signals that are clearly visible with FLI. The explanation given is not convincing about the superiority of PACT over FLI.

Reply: Please see Reviewer 5's Comment 4.

6. The authors never address nor attempt ventral imaging as suggested by Reviewer 4.

Reply: As suggested during the previous review round, we additionally provided ventral-view imaging by administering a higher concentration of ICG. The resulting signals observed in the cervical lymph nodes and nasal region were included in Supplementary Figure 7. This was added because the ventral image did not show detectable ICG signals when using the original 1 mM ICG concentration; therefore, we included this result to demonstrate that detection becomes feasible at higher concentrations. For further details regarding the molecular sensitivity of PACT in this context, please refer to our response to Comment 4.

Supplementary Figure 7. Dorsal and ventral PACT images after 12mM ICG injection.

Moderate-to-major concerns based on this reviewer's read-through:

1. Figure 3 is probably the most informative and novel figure of the paper. The description of this figure in the results should be re-written for conciseness. Several of the paragraphs are redundant. The bar charts are detracting from the rigor of the figure. Throughout the manuscript, figures should have individual animal information. With such small sample sizes throughout there is no reason to have bar chart plots.

Reply: *We thank the reviewer for the constructive comments. Following your suggestions, the bar charts in Figures 1 and 3 have been updated to include individual animal data points overlaid as dot plots, thereby providing full visualization of within-group variability and addressing concerns regarding sample size.*

Furthermore, we removed the redundant sentences in the Results section for Figure 3.

2. The intro is four pages long, yet the discussion is only two pages with a large portion devoted to rehashing results. Is there a way to rewrite to shorten the introduction, minimize extensive rehashing of results/bring attention to how PACT has both important new innovations and significant caveats in the discussion?

Reply: *Thank you for this insightful comment. We have revised the introduction by removing redundant explanations and streamlining background information to make it more concise. In the discussion, we added a clearer description of the strengths and limitations of PACT, particularly in comparison with fluorescence imaging. For details of these revisions, please refer to our response to Comment 4:*

Page 16, Line 302:

“To date, CSF circulation and lymphatic drainage have primarily been investigated using FLI, MRI, and PET. However, each technique has fundamental drawbacks: FLI lacks depth and 3-D quantitative information, MRI offers limited temporal resolution and requires lengthy acquisitions, and PET relies on radioactive tracers and lacks detailed anatomical context. In contrast, PACT is an emerging biomedical imaging modality that leverages its multiparametric capabilities and has been widely applied in preclinical mouse studies across diverse disease models, including cancer and stroke^{29,53}. In this study, we demonstrate for the first time that PACT can be utilized to monitor CSF dynamics and glymphatic-related function in rodent models, highlighting its potential as a next-generation platform for evaluating fluid transport in the central nervous system. Meanwhile, photoacoustic microscopy (PAM) is another imaging technique that utilizes the PA effect, similar to PACT, but focuses on micro-scale imaging as opposed to PACT’s macro-scale imaging. There have been studies employing PAM to explore multi-wavelength imaging of the dynamic processes of CSF in meningeal lymphatic vessels^{42,54}. These studies have contributed valuable data on the localization and movement of CSF. However, they have several limitations. This limitation fundamentally arises from PAM’s single-element scanning architecture, which acquires point-by-point A-line signals rather than simultaneously capturing 2D or 3D volumes. First, previous PAM studies primarily provided non-real-time imaging data, which limits the ability to dynamically monitor and quantify CSF flow under various physiological conditions. This limitation makes it difficult to capture rapid physiological changes in homeostasis. In other words, these studies lack the capability to dynamically monitor variations in CSF flow, which is crucial for understanding the glymphatic function. Second, previous research has predominantly focused on localized brain imaging, which restricts the understanding of systemic physiological interactions and cannot provide a comprehensive view of CSF distribution in major lymphatics on a macro scale. Consequently, while previous PAM studies have laid important groundwork for CSF imaging using the PA effect, there remains an opportunity to develop more holistic, real-time, and comprehensive insights into CSF dynamics and glymphatic function via PACT. This study demonstrates the advantages of PACT, a novel biomedical imaging technique for analyzing CSF dynamics and glymphatic function.

In this study, we visualized major lymphatic regions in key lymphatic regions including the subarachnoid space adjacent to the spinal cord and the CM using ICG as a CSF tracer, and it allows observation of CSF’s circulation.”

3. The references for the anesthesia effect (the authors use numbers 12 and 13) are incorrect, or, at minimum, not what the field cites for the effect of anesthesia on glymphatic function. More relevant citations should be included, if not used to replace these two.

Reply: *We appreciate the reviewer's comment. The citations regarding the anesthesia-related glymphatic effects have been updated accordingly.*

*[Ref. 48] Benveniste, H., Heerdt, P. M., Fontes, M., Rothman, D. L. & Volkow, N. D. Glymphatic System Function in Relation to Anesthesia and Sleep States. *Anesth Analg* 128, 747-758, doi:10.1213/ane.0000000000004069 (2019).*

However, by referencing these studies, we acknowledge that the effects of anesthesia on glymphatic function remain a subject of considerable debate.

- Xie et al., *Science* 2013, which first demonstrated enhanced glymphatic influx during sleep and under ketamine/xylazine anesthesia relative to wakefulness and reported differential effects of isoflurane.*
- Gakuba et al., *Theranostics* 2018, which challenged these findings by showing suppression of glymphatic activity under general anesthesia and sparked the ongoing debate regarding anesthesia dependency in the glymphatic field.*
- Dong et al., *Neurobiology of Disease* 2024, which provided new evidence that long-term isoflurane anesthesia impairs glymphatic function by disrupting AQP4 polarization and reducing the clearance of inflammatory proteins.*

*[R1] Xie, Lulu, et al. "Sleep drives metabolite clearance from the adult brain." *science* 342.6156 (2013): 373-377.*

*[R2] Gakuba, Clement, et al. "General anesthesia inhibits the activity of the "glymphatic system"." *Theranostics* 8.3 (2018): 710.*

*[R3] Dong, Rui, et al. "Long-term isoflurane anesthesia induces cognitive deficits via AQP4 depolarization mediated blunted glymphatic inflammatory proteins clearance." *Journal of Cerebral Blood Flow & Metabolism* 44.8 (2024): 1450-1466.*

These studies highlight that the effect of anesthesia on glymphatic transport remains an area of active discussion, and we fully recognize this controversy.

*Given space and scope considerations, and to avoid overloading the manuscript with mechanistic debate beyond our study's intent, we have cited the field-standard review by Benveniste et al., *Anesth Analg* 2019, which synthesizes the anesthesia–glymphatic literature comprehensively and is the most representative source for readers.*

Our study does not aim to resolve the physiological controversy itself; rather, it aims to show that PACT can detect the well-described differences in CSF movement under ketamine/xylazine versus isoflurane, demonstrating PACT's utility as an assessment tool. For further clarification, please refer to our response to Reviewer 5's Major Comment 2.

4. Reference 13 for the AD mouse model may not be the most applicable/suitable reference. Can the authors provide additional examples that are measuring perivascular flow and tracer movement within the brain instead of just the lymphatic system?

Reply: *Thank you for the valuable comment. We have added several key references properly.*

Page 13, Line 253:

“Compared to WT mice, AD mice showed impaired brain clearance due to the accumulation of A β , which is associated with likely disrupted lymphatic pathways and reduced CSF flow^{13,43-45}.”

[Ref. 43] Harrison, I. F. et al. Impaired glymphatic function and clearance of tau in an Alzheimer's disease model. Brain 143, 2576-2593 (2020).

[Ref. 44] Iliff, J. J. et al. A paravascular pathway facilitates CSF flow through the brain parenchyma and the clearance of interstitial solutes, including amyloid β . Science translational medicine 4, 147ra111-147ra111 (2012).

[Ref. 45] Pappolla, M. et al. Evidence for lymphatic A β clearance in Alzheimer's transgenic mice. Neurobiology of disease 71, 215-219 (2014).

Minor comments:

1. May want to double-check use of ISF acronyms in intro

Reply: *Thank you for the comment. We have carefully reviewed the manuscript and corrected all acronyms.*

Page 2, Line 39:

“Cerebrospinal fluid (CSF) both nourishes brain cells and clears the central nervous system (CNS) of metabolic waste and pathological proteins such as amyloid beta (A β) and tau, which cause neurodegenerative diseases. Because the brain lacks a general lymphatic system, waste is cleared via CSF circulation in the glymphatic system, a specialized perivascular network that promotes the exchange of CSF and interstitial fluid (ISF), facilitating the removal of metabolic waste and neurotoxic proteins. Passing through the subarachnoid space, CSF flows into the deep brain, penetrating the brain parenchyma and mixing with the ISF. The combined CSF/ISF fluid clears waste and drains via arachnoid granulations into the venous sinuses and via meningeal lymphatic vessels into the cervical lymph nodes. The meningeal lymphatic system disposes of brain and spinal cord waste by transporting ISF and CSF.”

2. The color of the arrows in Figure 1 may need to be more saturated.

Reply: *The arrows in Figure 1 have been updated with increased color saturation to improve visibility and clarity.*